



Manuscript prepared for Geosci. Instrum. Method. Data Syst.
with version 5.0 of the LaTeX class copernicus.cls.
Date: 28 November 2019

# Progress in managing the transition from the RS92 to the Vaisala RS41 as the operational radiosonde within the GCOS Reference Upper-Air Network

Ruud J. Dirksen[1], Greg E. Bodeker[2], Peter W. Thorne[3], Andrea Merlone[4], Tony Reale[5], Junhong Wang[6], Dale F. Hurst[7], Belay B. Demoz[8], Tom D. Gardiner[9], Bruce Ingleby[10], Michael Sommer[1], Christoph von Rohden[1], and Thierry Leblanc[11]

[1]GRUAN Lead Centre, Deutscher Wetterdienst, Meteorologisches Observatorium Lindenberg, Am Observatorium 12, 15848 Tauche/Lindenberg, Germany
[2]Bodeker Scientific, Alexandra, New Zealand
[3]Maynooth University, Maynooth, Ireland
[4]INRI, Turin, Italy
[5]NOAA/NESDIS, Washington DC, USA
[6]Department of Atmospheric and Environmental Sciences, State University of New York, Albany, USA
[7]NOAA Earth System Research Laboratory, Boulder, CO, USA
[8]Howard University, Washington D.C., USA
[9]National Physical Laboratory, Teddington, UK
[10]ECMWF, Reading, UK
[11]JPL, Pasadena, USA

*Correspondence to:* R. J. Dirksen (Ruud.Dirksen@dwd.de)

**Abstract.** This paper describes the GRUAN-wide approach to manage the transition from the Vaisala RS92 to the Vaisala RS41 as the operational radiosonde. The goal of the GCOS Reference Upper-Air Network (GRUAN) is to provide long-term high-quality reference observations of upper air Essential Climate Variables (ECVs) such as temperature and water vapor. With GRUAN

data being used for climate monitoring, it is vital that the change of measurement system does not introduce inhomogeneities in to the data record. The majority of the 27 GRUAN sites were launching the RS92 as their operational radiosonde, and following the end of production of the RS92 in the last quarter of 2017, most of these sites have now switched to the RS41. Such a large-scale change in instrumentation is unprecedented in the history of GRUAN and poses a challenge for the

network. Several measurement programmes have been initiated to characterize differences in biases, uncertainties and noise between the two radiosonde types. These include laboratory characterization of measurement errors, extensive twin sounding studies with RS92 and RS41 on the same balloon, and comparison with ancillary data. This integrated approach is commensurate with the GRUAN principles of traceability and deliberate redundancy. A two-year period of regular twin soundings

is recommended, and for sites that are not able to implement this burden sharing is employed, such that measurements at a certain site are considered representative of other sites with similar climatological characteristics. All data relevant to the RS92-RS41 transition are archived in a database that





will be accessible to the scientific community for external scrutiny. Furthermore, the knowledge and experience gained about GRUAN's RS92-RS41 transition will be extensively documented to ensure

traceability of the process. This documentation will benefit other networks in managing changes in their operational radiosonde systems.

Preliminary analysis of the laboratory experiments indicates that the manufacturer's calibration of the RS41's temperature and humidity sensors is more accurate than for the RS92; with uncertainties of $< 0.1$ K for the temperature and $< 1\%$ RH for the humidity sensor. A first analysis of 224 RS92-

RS41 twin soundings at Lindenberg Observatory show nighttime temperature differences $<0.1$ K. However, daytime temperature differences in the stratosphere increase steadily with altitude, with $T_{RS92\text{-}GDP.2}$ up to $0.6$ K higher than $T_{RS41}$ at $35$ km. $RH_{RS41}$ values are up to 8% higher, which is consistent with the analysis of satellite-radiosonde collocations.

## 1   Introduction

The Global Climate Observing System (GCOS) has instigated the GCOS Reference Upper-Air Network (GRUAN) (Bodeker et al., 2016) to perform reference quality measurements of upper-air Essential Climate Variables (ECVs) (Bojinski et al., 2014). The network consists of a range of national contributions of high-quality observing facilities undertaking observations in a systematically similar manner that assures traceability to System of Units (SI) or community accepted standards, with

full quantification of the uncertainties arising from every step in the processing chain. To date, the Vaisala RS92 and the Meisei RS-11G radiosondes are the only GRUAN data products currently being routinely produced and disseminated to the user community (Dirksen et al., 2014; Kizu et al., 2018). Several additional products, including those from other radiosonde models, ozone sonde and a variety of remote sensing techniques, are at various stages of maturity and should be avail-

able for analysis in the near future. Readers interested in further details of GRUAN, its development and its operation, are encouraged to read Bodeker et al. (2016) or to visit the GRUAN website at https://www.gruan.org. A map showing sites participating in GRUAN is shown in Figure 1. Currently, RS92 data from GRUAN sites are available via anonymous ftp from NOAA's National Centers for Environmental Information at ftp://ftp.ncdc.noaa.gov/pub/data/gruan/processing/level2/

RS92-GDP/version-002.

In the course of 2014, Vaisala announced the introduction of the RS41 as the RS92's successor, and that the production of the RS92 radiosonde would be terminated by the end of 2017. Before its end of production, Vaisala's RS92 radiosonde was also widely used outside of GRUAN, with a global market share of approximately 30% (including at least daily launches at sites on every

continent). Its performance was among the best of the commercially available radiosonde models (Nash et al., 2011). Until recently, the majority of the 27 GRUAN sites employed the RS92 (listed in Table 1 and shown in Figure 1), which effectively made the RS92 GRUAN's backbone in terms of





upper-air sounding. Any change in instrumentation in a GOS (Global Observing System) network not only presents potential data continuity concerns, but it will also pose a challenge to a far broader

community of users. Radiosounding data form a key input to NWP systems and human forecasts such that any difference in performance has potentially large impacts. The challenge is to ensure continuity of operations without negative scientific or financial ramifications. The manual on the GOS states that "Changes of bias caused by changes in instrumentation should be evaluated by a sufficient period of observation (perhaps as much as a year) or by making use of the results of

instrument intercomparisons made at designated test sites" (Section 2.2.2.13 of WMO, 2017).

One of the key potential benefits of a tiered networks design (Bodeker et al., 2016) is the dissemination of information derived from a subset of top-tier reference quality sites down to the geographically broader lower-tier network sites. Figure 2 schematically depicts a three-tiered upper-air observing system architecture, with a 30-40 station GRUAN network providing reference observa-

tions for more extensive networks such as the GCOS upper-air network (GUAN). Reference network sites serve as the long-term anchor points that comprehensively characterize the atmospheric column with the highest quality measurements currently feasible. The base of the system is the entire global upper-air observing system, serving a wide variety of purposes, primarily weather prediction, and including the operational radiosonde network, aircraft and satellite observations, etc., and embracing

model-assimilated upper-air datasets and reanalyses. The 177-station (as of March 2019) GUAN is a subset of the operational radiosonde network that, in the late 1990s, committed to long-term, consistent observations, but does not deploy any special instruments for high-quality climate observations as GRUAN does.

In the case of the transition from RS92 to other radiosonde types, the lessons learnt from GRUAN

activities to manage the transition may benefit those GUAN sites faced with the same challenge as well as other sonde stations from the remainder of the GOS. Furthermore, by undertaking an intensive characterization of the transition, GRUAN can assist not just the climate community but also other communities such as numerical weather prediction (NWP)/forecasting through active dissemination of the resulting analyses of any effects of the transition. Such an approach requires visibility

of the change management process and associated analysis results by the community through transparency of the GRUAN transition results, which will provide the confidence required to encourage uptake of the results.

This paper is a first step in this process and serves to highlight the activities undertaken to date and planned future activities that GRUAN (including the sites, Lead Centre, working group and

various task teams) intends to carry out in support of the change management. Community feedback, which may strengthen the planned activities, is strongly encouraged. In particular we would eagerly welcome support in:

- Provision of available data from RS92 + RS41 twin launches (i.e., both on the same balloon) from non-GRUAN sites to expand the geographical characterization of any biases between


RS92 radiosondes and their successors.

– Offers of assistance from the expert user community in the synthesis of twin launch data from
multiple sites.

– Suggestions of additional analyses that could be performed.

The remainder of this paper outlines various aspects of the change strategy, such as a network-wide
approach including burden-sharing, the application of ancillary data, the metrological perspective
on the change, the role of documentation and the creation of a scientific database which stores
all measurement data which are relevant to the change. Furthermore, it reports on the progress to
date and current plans to diagnose biases between RS92 and RS41 radiosondes and provides some
initial results based on analysis of 224 twin soundings that have been performed at the Lindenberg
Observatory. The data of the laboratory experiments that were used for the plots in this paper are
stored in a permanent repository with digital object identifier (DOI) 10.5676/GRUAN/dpkg-2019-1
(Dirksen and Sommer, 2019).

## 2   The challenge

For any long-term measurement series, inevitably, the challenge of change management is certain
to arise. This could be either through choice when improved or more efficient means of making
the measurements become available, or through necessity as an instrument and its manufacturer
support become unavailable. Regardless of whether by choice or necessity, the central challenge
is to manage the transition in such a manner that the adverse effects on a long-term measurement
series' continuity and homogeneity are minimized and any uncertainties propagated by the transition
are well understood. Sufficient data, and associated metadata, are required both to perform an initial
analysis and to permit future reprocessing and reanalysis of the data.

Several National Weather Service (NWS) organizations implement change management strate-
gies that are executed within their organization. An example of such an activity is the Radiosonde
Replacement Plan for the NOAA/NWS (see Peterson and Durre (2004)).

While GRUAN does its best to avoid unnecessary changes in its operating protocols, it is not im-
mune from the challenges of change management. Change management has been a key component
of the network's goal to constantly strive to make the best possible measurements of the relevant
ECVs. As such, GRUAN has never been envisioned as a network where the underlying instrumen-
tation shall remain forever static. Prior to this forced transition away from RS92 sondes, only one
GRUAN site (Tateno, Japan) had undergone a change in their radiosounding system, namely the
switch from the Meisei RS2-91 to Vaisala's RS92 radiosonde in 2009-2010 (Kobayashi et al., 2012).
This site subsequently switched from RS92 to Meisei RS-11G in 2015-2017.

In case of the RS2-91 to RS92 switch, the change management included a series of twin launches





during all four seasons. The study highlighted the importance of undertaking a sustained programme
(i.e. >1 year) of coincident soundings by old and new instrumentation to understand any seasonality
of biases. Seasonally dependent biases may arise from changes in the measured ECVs and/or annual
cycles of covariates such as radiation effects which are particularly important at those sites where
launches systematically occur at or near dusk/dawn. The second switch, RS92 to RS-11G, involved
52 twin soundings over a period of two years (Kobayashi et al., 2019).

There are some crucial distinctions between this precursor analysis at Tateno and the current
GRUAN-wide transition from RS92 to RS41 radiosondes. For Tateno:

– At the time, neither the original nor the replacement sonde models had GRUAN data products
being processed and provided routinely to the user community.

– The change related to a single instrument at a single site.

– The update arose from a choice by the site to change instrumentation such that the timetable
could be altered as necessary.

Although the matter of broader change management pertaining to simultaneous instrument transi-
tions at multiple sites has been informally discussed on various occasions, e.g. during GRUAN's an-
nual Implementation and Coordination Meetings (ICMs), prior to the RS92 cessation of production,
there existed no formal plan for managing such a wide-spread change. This large-scale transition
poses a major challenge for GRUAN as a reference network because it must not compromise the
continuity, quality, and homogeneity of the data records.

This change of the operational radiosonde at the majority of the GRUAN sites is unprecedented
in the history of GRUAN. A network-wide challenge requires a facilitated and coordinated solution
if the change is to be successful, and if GRUAN is to succeed. The fundamental challenge is thus
to design and deliver a GRUAN-wide strategy for managing and coordinating the near-simultaneous
changes of the operational radiosonde at many sites. This strategy should include all aspects of the
change management including inter-alia:

– network coordination to share the burden,

– the necessary roles of ancillary measurements coincident with the sonde measurements,

– instrument characterization in the laboratory and field under a wide range of conditions,

– scientific analysis,

– technical documentation, and scientific publication,

– dissemination of results to GRUAN and the broader GOS through GCOS/ World Meteorolog-
ical Organization (WMO) Integrated Global Observing System (WIGOS) mechanisms,





– sharing information with agencies conducting Near Real Time (NRT) data applications to aid their managing of the change.

Since GRUAN always seeks to promote competition in the marketplace, sites were encouraged to consider all available options for how to proceed with their transition from the RS92, including changing to sondes produced by other manufacturers. Despite the independent decision-making procedure, each GRUAN site that launched RS92 sondes has transitioned to the RS41. It is important to stress that these decisions arose from the individual GRUAN sites and not from the network management. The consequence is that the network must transition between two instrument models from the same manufacturer. If some sites had chosen to switch to radiosondes from different manufacturers, GRUAN would have been required to develop a very different change management programme than the one described here.

Two of GRUAN's strengths are its ability to call on expertise from across the network to tackle such challenges and its ability to distribute required actions among the sites to share the burden. Currently, specialists from various fields of expertise within GRUAN are engaged in addressing the above mentioned points. This not only advantages the affected sites and GRUAN as a whole, but also helps other observational networks, such as e.g. GUAN, in managing the same transition.

## 3   Change management

Proper management of the change of a measurement system requires determining all relevant differences between both systems prior to the transition. Typically this means organizing a period of observational system overlap as well as laboratory-based characterization of the differences between the instruments. While the specialized facilities to perform extensive laboratory testing are not available at each site, there is no impediment to sites performing real-world intercomparisons like twin soundings, although the costs of extra receiving systems and sondes may pose limits on the number of flights that can be performed.

It is essential to quantify biases between the new and the old instruments as well as changes in calibration/measurement errors and uncertainties. These attributes may have complex interactions with covariates which complicate the quantification of the effects of the change. One example of such a measurement error, the solar radiation-induced temperature bias, will vary with altitude, season and geographical location, because it depends on the ambient pressure, solar elevation angle and radiation intensity (Dirksen et al., 2014). The full range of sources of RS92 uncertainties that may have complex spatio-temporal characteristics, including ventilation, sensor orientation, and pre-launch calibration, have been described in detail by Dirksen et al. (2014). These uncertainties will also vary with location due their dependence on solar elevation angle, cloudiness, and winds. Once the biases have been identified and corrected for, and the uncertainties have been determined, the data ($m_1$ and $m_2$) from both measurement systems should be consistent, meaning that the agreement





criteria set out in Equation 1 are met with factor $k = 2$ (Immler et al., 2010). Another way of saying this is that the measurement data consistently lie within each other's uncertainty coverage factors ($u_1$ and $u_2$) after accounting for any effects of non-coincidence.

$$|m_1 - m_2| < k \cdot \sqrt{u_1^2 + u_2^2} \qquad (1)$$

Verifying this consistency requires a sufficient population of coincident measurements to determine in a statistically robust manner that the data satisfy this condition. Neither the GRUAN Manual (WMO, 2013a) nor the GRUAN Guide (WMO, 2013b) provide a clear requirement for the duration and intensity (measurement frequency) of an intercomparison study because it is inherently dependent on instrument type and measurement principle. However, as mentioned in Section 1, the manual

on the GOS suggests a one-year intercomparison period. Different instruments measure different ECVs in very distinct manners. Some are episodic remote sensing, some are continuous remote sensing, others are periodic in-situ profiles. For each measurement type a different 'side-by-side' operation strategy would be required, which depends on (at a minimum): the instrument characteristics, variability in the target measurand, and cost and logistical considerations. Kobayashi et al.

(2012) suggest that a total of 120 twin soundings spread across the seasonal cycle, at a given location, would be more than sufficient to characterize the effects of a change in radiosonde instrumentation, although the study did not consider the metrological quantification aspects that are necessary in the current case. Hence, for radiosondes, the GRUAN Lead Centre recommended that sites perform weekly, or with a bi-weekly interval, twin soundings for a period of two years. The two-year period

ensures that seasonality is better probed, and a sounding interval of a week instead of a day mitigates the additional operational costs. The twin soundings should be equally distributed between day and night soundings.

Coordination with satellite overpass is highly encouraged, with particular emphasis on targeting times when a GNSS-RO and polar orbiter overpass may occur in the site's proximity. The

sonde-satellite data comparison would also serve as an additional station-to-station transfer-check of consistency of the results. This is further discussed in Section 7.

Laboratory testing is used to characterize the radiosonde's sensors, as is done to establish a GRUAN data product for each radiosonde model. The results of the laboratory tests for the RS92-RS41 transition are being employed in developing a GRUAN data product for the RS41. The pa-

rameters tested include: radiation error for the temperature and humidity sensors, sensor calibration accuracy, and response time-lag and hysteresis effects of the humidity sensor. The Lead Centre facility, hosted by DWD at the Lindenberg Observatory, has access to a broad range of laboratory-based facilities suitable for characterizing radiosonde performance. These facilities, shown in Figure 3, were used to characterize the RS92 instrument as described in Dirksen et al. (2014) and include:

225        – standard humidity chambers (SHCs) validate RH sensor calibration,

    – radiation test chambers to investigate the solar-induced temperature errors and dry biases,



– climate chamber operating between –75 °C and +20 °C for sensor response time-lag testing.

Section 5 gives a more detailed discussion of the application of these facilities to the characterization of radiosondes, together with an overview of preliminary results from these laboratory tests.

## 4  Network coordination to share the burden

### 4.1  Introduction

From a GRUAN perspective, the collective transition from the RS92 to the RS41 radiosonde encourages approaches to burden sharing this change management. Here, burden sharing means that selected sites, each representative of a different climate region (e.g., tropical, mid-latitude, polar) perform prolonged intercomparison studies, while climatologically similar sites are not necessarily required to perform such campaigns. This reduces the logistical burden on the site and the cost of additional sondes and receiving systems. Other forms of burden sharing could be implemented. For example, climatically equivalent sites can divide the workload such that one site conducts daytime twin flights while another site performs the nighttime twin flights. A similar division of labor could be agreed upon to balance sampling of different seasons.

This approach to burden sharing will promote quality over quantity, as it is envisaged that some sites will opt only for a limited intercomparison period, or perhaps only a short campaign-like effort. On their own, these limited intercomparison efforts would be insufficient to properly investigate the seasonality of the differences between the old and the new sounding systems. But as a network, the sum of these small contributions becomes substantial. Table 1 lists the GRUAN sites that switched from RS92 to RS41 and, for those that decided to intercompare the two sondes, the actual or proposed duration and frequency of their twin sounding programmes. Based on these results, the mid-latitudes climate regions in both hemispheres and the Arctic are covered by long-term intercomparison efforts. Clearly lacking are intercomparison efforts in the tropics and the Antarctic, a consequence of there being no GRUAN sites with an established operational RS92 measurement programme in those regions. However, there is one (candidate) GRUAN site in the tropics that did employ the RS92 (the BoM-operated site in Darwin, Australia). This site has also decided to switch to RS41 and has performed a condensed intercomparison programme, which partially fills the gap in the tropics.

In addition to the efforts by GRUAN sites, several non-GRUAN sites have also performed intercomparison campaigns as a way to manage their RS92-RS41 transition. These include the UK Met Office comparisons in St Helena and in Rothera in Antarctica. Some of these sites have agreed to share their twin sounding data with GRUAN, providing critical data in regions that are not well-served by GRUAN sites. These sites are listed in Table 2. Furthermore, the GRUAN Lead Centre was involved in the StratoClim campaigns of 2016 and 2017 (India and Nepal, respectively Brunamonti et al., 2018; Rex, 2014). During these campaigns, that took place in July-August during the





Indian Monsoon, RS92-RS41 twin soundings were performed to investigate the differences between
both systems under these particular meteorological conditions.

All data from the intercomparisons listed in Tables 1 and 2 will be made available to the scientific
community via the GRUAN data servers, as discussed further in Section 8.3.

### 4.2   Support equipment

To support data telemetry for RS41 radiosonde measurements at GRUAN sites, the Lead Centre has
a spare, fully-equipped radiosounding receiving system that can be temporarily loaned to sites that
cannot afford to purchase a second receiving station. This fully functioning system consists of an
antenna, an MW41 receiving system, and a compact Vaisala WXT weather station for collecting
metadata surface observations at the time of the launch. In addition to this hardware, an SHC can be
loaned for performing manufacturer-independent pre-launch checks in a 100 %RH environment, as
discussed by Dirksen et al. (2014).

Furthermore, Vaisala has several MW41 systems on hand that can be loaned to sites that wish to
conduct short to medium-term RS92-RS41 intercomparison campaigns. Various GRUAN sites have
used this option to manage their transition, and loaned systems have also been employed during
campaigns, such as the StratoClim campaigns, and the CONCIRTO campaign on Reunion Island in
2019.

### 5   Laboratory characterization

### 5.1   Introduction

The laboratory facilities at the Lead Centre/Lindenberg Observatory, photographically presented in
Figure 3, are being used for extensive testing to characterize the measurement errors and uncertain-
ties of the RS41 and the RS92, an activity which is essential for the development of a GRUAN data
products for both radiosondes. The tests primarily focus on the error sources which are known to be
dominant for radiosondes, i.e. the solar radiation heating of the temperature and humidity sensors,
the response time-lag, and the accuracy and reproducibility of the humidity sensor's calibration.

Radiation error tests are performed in an adapted SHC at pressures between ambient and $3\,hPa$
(see Dirksen et al. (2014) for a description of the radiation tests and their configuration) as well
as in a newly developed system that allows for improved ventilation and illumination. Preliminary
radiation tests were performed on RS41 radiosondes from 2014 through 2019, and further tests are
foreseen for 2020. The results indicate that the temperature sensor of the RS41 radiosonde is less
susceptible to heating by solar radiation than that of the RS92. However, these results apply to raw
(uncorrected) measurement data and it is not possible to draw direct conclusions on the resulting
temperature bias between the Vaisala-processed data products of RS92 and RS41 since in the data
processing of both sondes different corrections are applied for the radiative heating.





The calibration accuracy and hysteresis effects of the humidity sensor have been investigated by placing the radiosonde's sensor boom in an SHC having a stable, well-defined RH between 0 % and 100 %. The stable RH environment inside each SHC is achieved using one of the saline mixtures listed in Table 4. In addition, each SHC is equipped with a Pt100 reference thermometer which tests

the calibration accuracy of the radiosonde temperature sensor.

The response time-lag of the humidity sensor is determined by measuring its reaction to a stepwise change in the humidity of the airflow, while keeping the temperature of the airflow constant. The time-lag effect becomes significant at temperatures below –40°C, the point at which the response time of the humidity sensor starts to exceed 10 s (Miloshevich et al., 2004; Dirksen et al., 2014). The

time-lag tests are performed in a climate chamber which can reach temperatures as low as –75°C. Table 3 summarizes the laboratory experiments that have been performed to date to characterize the RS41 radiosonde.

The lab-based characterization results will be included in the scientific database holding all data that are relevant to the RS92 to RS41 transition (discussed in Section 8.3).

**5.2   Results of the laboratory characterization**

To assess the calibration accuracy of the humidity and temperature sensors, more than 150 RS41 radiosondes from various production batches were tested in SHCs at the relative humidities listed in Table 4. In a typical experiment, each RS41 was sequentially placed in a series of six SHCs with increasing relative humidity, from 0 to 100 %, then back through the sequence of drier SHCs to

0 %RH. This sequence also allows assessment of hysteresis of the humidity sensor. At each humidity level, the radiosonde was immersed in the SHC for approximately 4 minutes while its readings were recorded. The air inside each SHC was circulated at 5 m/s by a fan, and the temperatures of the air and saline solution inside the SHCs were measured by Pt100 reference thermometers. For some salts both air and solution temperatures are needed to accurately determine the relative humidity in

the SHC, since the humidity over the reference salt depends on both quantities.

Figure 4 shows that the majority of the temperature measurements by the RS41 are within ±0.5 K of the reference temperature. Although the tail of the distribution extends to 1 K (not shown), only a few measurements show differences beyond 0.5 K. The mode of the distribution indicates a bias of -0.025 K, while from its width it can be inferred that the calibration uncertainty is smaller than

0.1 K. The histogram of temperature differences between the RS41 and the Pt100 reference is not Gaussian, something that cannot yet be explained but will be the subject of further investigation.

Figure 5 shows that all humidity measurements by the RS41 are within 2 %RH of the reference RH value with only a small fraction of the measurements showing differences larger than 1 %RH, which means that the uncertainty of the humidity calibration is smaller than 1 %RH. The histogram of RH

differences between the RS41 and the reference RH value has a pronounced peak around 0 %RH, due to the fact that the calibration errors are dependent on RH. At low RH the differences, and their





spread, between the reference and the RS41 values are small, whereas at high RH the differences
and spread are larger. As a result, all observed calibration errors at 0 %RH humidity cluster around
0 %, thereby over-representing these values in the distribution.

Radiation tests were conducted in the modified SHC as described by Dirksen et al. (2014). Mea-
surements were performed at various settings within the ranges:

- pressure between 3 hPa and ambient,

- irradiance from 200 to 1000 W/m$^2$,

- ventilation speed of either 2.5 or 5 m/s,

- illumination times of 1-4 minutes.

    The illumination times depended on pressure, with longer exposures required for lower pressure
environments to reach equilibrium. Figure 6 shows similar heating of the RS92 and RS41 tempera-
ture sensors at 300 hPa (0.15 K), but at 3 hPa (Figure 7) the RS41 (1.4 K) heats only half as much
as the RS92 (2.8 K). Furthermore, at 3 hPa, the RS41 sensor has reached equilibrium after approx-
imately 30 s of illumination, whereas after 4 minutes of illumination the RS92 still hasn't reached
equilibrium. The investigation of the RS41's radiation error continues, including experiments with
the newly developed laboratory radiation system, and the full analysis of these tests will be reported
in a separate paper.

## 6   Metrology

A fundamental metrological principle stipulates that replacing one operational instrument with an-
other should pose no problem provided that the results from both instruments are fully traceable to
SI standards. Consequently, the new instrument could (almost) instantaneously be included in the
traceability chain without the need for parallel testing or comparison with the replaced device. In
practice, this idealized concept can rarely be adopted, even in primary metrology laboratories or
in National Metrology Institutes. The problem is that different instruments or sensors may show
different responses to external environmental factors.

    Concerning radiosondes, the sensors, especially for humidity and temperature, may during a
sounding be exposed to unavoidable atmospheric or ascent-related effects. Some of these effects
cannot fully be included in the realization of the controlled laboratory conditions which are pro-
vided during the preceding metrological instrument characterization and calibration procedures.
Consequently, differences in the responses of sensors from different sonde models may still exist
during ascents. An example is the warm bias of the radiosonde's temperature sensor caused by solar
radiation.

    For this reason it is essential to identify and quantify these differences between the old and the new
measurement system at each time of a sonde replacement not only by laboratory work but also by



comparison flights where the instruments to be compared are mounted under the same balloon. The same should also be done when there have been significant changes in the design, sensor technology, or operation of an actual instrument.

An adequate characterization of the measurement uncertainties for entire radiosonde profiles includes the following components:

- Calibration uncertainty, given by the manufacturer (see product data sheets for Vaisala RS92 and RS41 available the manufacturer's web site, https://www.vaisala.com). The sensors are calibrated in Vaisala's CAL4 calibration facility (Vaisala, 2002) that contains PTU (pressure, temperature, and humidity) reference sensors, routinely re-calibrated against National Institute of Science and Technology of the United States of America (NIST)-traceable standards (for pressure and temperature) and the Finnish Centre for Metrology and Accreditation (Mittatekniikan keskus – MIKES, which has become part of VTT Technical Research Centre of Finland Ltd and is now officially known as VTT-MIKES) for humidity. Note that for GRUAN the uncertainties of the calibration curves are important, not so much the information about the overall uncertainties in soundings which are related to the manufacturer-provided data product,

- Uncertainties from GRUAN-prescribed pre-flight ground check procedures,

- Uncertainties estimated during GRUAN post-flight processing, including uncertainties of corrections to the measured raw data for systematic effects which are mainly derived from laboratory tests.

Disclosed differences found in instrument comparison flights indicate that some systematic effects have not yet been correctly assessed or even identified. To assure consistency of the measurement results of different instruments, the differences are to be evaluated in terms of possible corrections, or, if this is not possible, the uncertainty budget is to be extended properly to account for them. It is important to arrange the comparisons network-wide in a coordinated manner, covering different time scales and locations. This is to ensure that as much aspects as possible are covered which may introduce systematics such as latitude, climate, environmental and technical conditions before and during launch, local specifics in the sounding procedures or setups.

An advantage of performing dual launches of RS92 and RS41 radiosondes is that some of the uncertainties related to the change process are moved from absolute evaluations to relative ones, making second order contributions to the uncertainty budget limited, if not negligible. Second order contributions are uncertainties in the quantification of any environmental factor that affects the measurement of the primary quantity, like the intensity of solar radiation impinging on the temperature sensor.

The investigation of external environmental factors on the measurement uncertainty of meteorological instruments is one of the key elements of the MeteoMet project (Merlone et al., 2015,





2018). MeteoMet aimed to link the metrological and meteorological communities and to apply robust metrological methods for retrieving traceable reference-quality meteorological observations that can be used for climate monitoring. For this purpose, MeteoMet advocates the application of climatic chambers which create environments of controlled temperature, pressure, and humidity settings, similar to the one at Lindenberg Observatory (Section 5).

Besides laboratory facility, transportable calibration chambers have been developed and involved in a field campaign at the Ny-Ålesund GRUAN site, for the calibration of ground based pre-launch check sensors for pressure and temperature (Musacchio et al., 2015). The novelty of the system was its capability to generate pressure and temperature independently, thus making the calibration curves inclusive of mutual factors of influence (mainly temperature on barometers). Measurement results become more accurate, being the calibration curves (or planes) more representative of the field conditions met also in such extreme polar environment.

## 7 Ancillary measurements

### 7.1 Introduction

Many GRUAN sites employ the principle of deliberate redundancy by simultaneously measuring a specific atmospheric parameter using different techniques. Examples include the use of remote sensing techniques such as Global Navigation Satellite System (GNSS), lidar, microwave radiometer (MWR), or Fourier-Transform Infrared Spectrometry (FTIR) in addition to the regular radiosonde flights. Ground-based observations, together with collocated satellite-based measurements, are referred to as ancillary data. These ancillary data add significant to the analysis of the twin-launch data, because they form an independent source of data that is not affected in the same way by the error sources that are typical for radiosoundings and therefore can be used to validate the radiosonde intercomparison data. Furthermore, within one orbit and among a limited number of consecutive orbits, satellites provide a consistent background on a global scale. However, the long-term calibration drifts and retrieval errors of space-borne instruments must be taken into account when comparing long-term data sets of coincident radiosoundings and satellite overpasses.

The GRUAN Lead Centre, in cooperation with the GRUAN Task Team Ancillary Measurements, is working with each GRUAN site to establish respective ancillary measurement data streams and ascertain which of these streams contain relevant data that could be used to support the RS92 to RS41 transition. A key goal is to establish scheduling and sampling protocols to provide ancillary information that is spatially and temporally synchronized and internally consistent (Equation 1). Protocols are being developed and deployed to ensure that such data are submitted to the scientific database (see Section 8.3) and tagged as ancillary information to facilitate future analyses.

The use of ancillary data from GRUAN sites in such a manner is currently in an early stage of development. Note that there are not yet officially certified GRUAN data products for the ancillary





measurements, although some like the GNSS-based total water vapor column product is likely to
be certified in the near future (Ning et al., 2016). Protocols for the development and the delivery of
geophysical profiles from the various remote sensing techniques are being finalized.

The National Oceanic and Atmospheric Administration (NOAA) Products Validation System
(NPROVS) (Reale et al., 2012) facility can associate satellite measurements with any tagged GRUAN
observation data within a given collocation window. This system can also process profile data from
coincident ancillary measurements, including the associated uncertainties. This includes redundancy
testing (Immler et al., 2010) and integration of consistent ancillary profiles into Site Atmospheric

State Atmospheric State Best Estimates (SASBE) (Tobin et al., 2006) as recommended by the Lead Centre. However,
the processing, packaging and use of the ancillary data in a spatially and temporally coherent manner
is a complicated task that remains under discussion.

The currently proposed plan is that the Lead Centre will identify each twin launch for the NOAA
NPROVS team, including site-specific ancillary data, once coincident measurement strategies and

data streams are established. The NPROVS team will append level-2 (geophysical profile) data
from satellites with special emphasis on those targeted for satellite overpasses and provide routine
monitoring, analysis and distribution.

## 7.2    Scheduling

In addition to ground-based ancillary measurements such as GNSS, lidar, MWR, and FTIR, satellite-

based observations also present a valuable and abundant source of additional observations for com-
parisons with radiosounding data. To maximize their potential for scientific exploitation, the ra-
diosonde launches should be scheduled to be coincident with satellite overpasses and/or the occur-
rence of GNSS-radio occultations (GNSS-RO) over the site. This would maximize the informational
content available from instrumentation both on-site and arising from satellite-based capabilities.

A joint service between GRUAN and European Organisation for the Exploitation of Meteorolog-
ical Satellites (EUMETSAT) provides predictions of these overpasses, including so-called "golden
overpasses" where the measurements of a polar orbiter, such as MetOp-A or MetOp-B, are spatially
(distance $< 200\,\mathrm{km}$) and temporally (within 30 minutes) coincident with a GNSS-RO measurement.
In addition, at GRUAN sites where ground-based ancillary measurements are also being made, there

could be multiple redundant observations of target ECVs to enable better exploitation; see Sec-
tion 7.1 for further discussion.

In the most basic context, satellites provide a consistent background or traveling calibration stan-
dard on a global scale to further interpret twin radiosonde results at a given site. Potential benefits
include the identification of possible site bias and whether a given set of twin results indicate an

improved (or degraded) representation of the actual atmospheric state. The additional information
from satellite-synchronized launches can potentially minimize the number of twin launches required,
reducing costs to the sites and maximizing opportunities for subsequent exploitation by the broader





science community for myriad applications on a global scale.

Examples of the potential benefits gained from redundant observations (Figure 8) were compiled
using the NPROVS (Reale et al., 2012).

Between September 2015 and November 2016, 58 RS92-RS41 twin soundings were performed
at the GRUAN site in Lauder, New Zealand. Among these, 15 were scheduled to coincide with
overpasses of EUMETSAT's MetOp-B satellite that hosts the Infrared Atmospheric Sounding Inter-
ferometer (IASI) and the Advanced Microwave Sounding Unit (AMSU).

The comparisons (Figure 8) show that the wet bias in the upper troposphere between satellite and
European Centre for Medium Range Weather Forecast (ECMWF) profiles and the Vaisala-processed
radiosonde data is smaller for the RS41 radiosonde than for the RS92 radiosonde, meaning that the
RS41 consistently reports higher RH values in the upper troposphere than the RS92. Both over
Lauder and Europe, the water vapor levels measured by the RS41 are up to 10% higher than for
the RS92. These results are consistent with Sun et al. (2016) who found that the bias of water vapor
mixing ratio in the upper troposphere (UT) from satellite observations and radiosonde measurements
(from GRUAN and non-GRUAN sites) was at least 10% smaller for the Vaisala RS41 compared to
RS92. This study was based on 6 month global samples of conventional RS41 and RS92 radiosondes
collocated with spaceborne infrared and microwave soundings from NOAA's Suomi National Polar-
orbiting Partnership (Suomi-NPP) satellite.

The decreasing differences between the radiosonde and both the satellite and the model data is
tentatively interpreted as the RS41 providing better RH measurements in the upper troposphere than
the RS92. However, validation by additional, independent measurements is needed to substantiate
this.

Figure 8 also shows that the bias between radiosonde and satellite/model data in the UT is up to
a factor two smaller for Europe than for Lauder. In addition, the shape of the difference profiles for
Lauder and Europe are different and analysis of this discrepancy is ongoing.

The finding that the RS41 measures higher humidity values in the UT than the RS92 is also
consistent with results for RS92-RS41 twin soundings that were performed at the GRUAN site in
Lamont/Southern Great Plains (SGP) in Oklahoma, USA, under management of ARM (Jensen et al.,
2016). However, these soundings were not performed in coincidence with satellite overpasses, so
comparisons with satellite data are not possible.

The collocated observations also permit assessment of calculated radiances, derived from ra-
diosonde profiles using radiative transfer models, versus observed satellite radiances. Calbet et al.
(2017) used this method to evaluate RS92 data, and when applying the same method to RS41 data it
can provide additional information on the differences between RS92 and RS41.

In summary, scheduling and targeting RS41-RS92 twin soundings with satellite overpass brings
more systems into the comparison and can enhance the transition analysis and provides more robust
physical interpretations of results under a wider variety of atmospheric conditions and locations.



The NPROVS programme provides routine coincidences of global conventional (including GUAN) and GRUAN reference radiosondes with environmental satellite observations (including GNSS-RO) within 6 hours and 250 km. This spatiotemporal criterion, taken from Reale et al. (2012), is a compromise between representativeness and sample size. This establishes a powerful baseline data set which can then be sub-sampled for detailed analysis depending on specific requirements for syn-

chronicity, cloudiness, etc.

## 8  Scientific analysis and publication

### 8.1  Introduction

The wide range of research activities investigating the RS92-RS41 transition that are outlined in this paper will result in a substantive, and valuable, data archive.

The analysis of this data archive will be done from various perspectives by scientists with different areas of expertise. Their results will need to be shared with the atmospheric science community. Table 5 lists a preliminary allocation of research analysis tasks and their principal investigators. With this multi-disciplinary approach to the analysis of the data it is anticipated that the differences between the RS41 and RS92 radiosondes will be well understood and that inhomogeneities in their

combined long-term data records can be minimized.

The list in Table 5 is incomplete and will be expanded as needs and requirements become clearer. The aim is to publish several distinct papers that describe the results. As further outlined in Section 9, we strongly welcome engagement by the broader atmospheric science community to analyze and publish the results.

### 8.2  Preliminary results

Up to September 2019, approximately 1500 RS92-RS41 twin soundings have been performed within GRUAN. A comprehensive analysis of this extensive data set is still ongoing, but as an example of this larger effort we present the preliminary results of the twin soundings performed at Lindenberg. According to Table 1, more than 400 twin soundings were performed in Lindenberg between Decem-

ber 2014 and July 2019. A subset of 224 twin soundings was available for the current analysis. The majority of these were performed with a payload consisting of an RS92 and RS41 radiosonde only, but a substantial number were performed with the RS92 and RS41 as part of an expanded scientific payload, consisting of additional instruments such as the Cryogenic Frostpoint Hygrometer (CFH) and Electrochemical Concentration Cell (ECC) ozonesonde. For the twin soundings, the payload

rig was configured consistent with the recommendations given in GRUAN-TN-7 (von Rohden et al., 2016), such that the radiosondes were attached with a 80 cm long string to each end of a 1.5 m long rod, ensuring free rotational movement of the radiosondes and minimizing potential contamination by water evaporating from the rod. In the analysis, GRUAN-processed RS92 profiles (RS92-GDP.2



Dirksen et al., 2014) are used whereas the RS41 data are processed by the Vaisala MW41 system.

Figure 9 shows the profile for such a twin sounding performed during daytime. The difference plot (right-hand panel) shows that up to the tropopause (in this particular case at approximately 14 km) both sondes capture the variations and structures in the humidity profile equally well, and the RS41 reports slightly higher (up to 2% RH) humidity values than the RS92. Above the tropopause, the RS41 reports lower RH values than RS92, which is attributed to the shorter time-lag of the

RS41 humidity sensor that is better able to capture the steep negative gradient in water vapor at the tropopause.

  The plots in Figure 10 (left-hand panel) show that for nighttime measurements the absolute temperature differences between the two sonde models are generally smaller than 0.05 K up to 30 km altitude with the RS92 (GRUAN-processed data) reporting slightly higher temperatures than RS41

(Vaisala-processed data). Above 30 km, $T_{RS41}$ increasingly exceeds $T_{RS92-GDP.2}$, by 0.1 K at 35 km. This indicates differences in the corrections for radiative cooling at the top of the profile for each radiosonde type. The right-hand panel of Figure 10 shows that the temperature differences for daytime measurements in the troposphere are smaller than 0.1 K, with $T_{RS92-GDP.2}$ larger than $T_{RS41}$. Above the tropopause this temperature difference gradually increases with altitude to approximately 0.6 K

at 35 km.

  Figure 11 shows that the tropospheric humidity values in the Vaisala-processed RS41 data are on average up to 5% higher at night and up to 10% higher during daytime. For the daytime measurements (right-hand panel) the relative differences increase with altitude, starting with a mean difference of 2.5% at the surface and reaching approximately 8% at 10 km. This observed higher

$RH_{RS41}$ in the Lindenberg twin soundings is consistent with the results presented in Section 7.2 and in Figure 8.

  More detailed and elaborate analyses of the differences between RS92 and RS41, including twin soundings from other (GRUAN) sites, will be performed in subsequent studies. There is already a considerable amount of data available, as is summarized in Tables 1 and 2. These studies will

investigate in detail the influence of geographical and climatological effects, such as solar elevation angle, clouds, and winds, on the RS92-RS41 differences.

### 8.3   Scientific database

A dedicated database, containing all data pertaining to the RS92-RS41 transition has been created and will be maintained. This database will be given its own digital object identifier (doi) and be pre-

served over the long-term. The purpose of the database is to have all relevant data for the transition available at a centralized location, thereby serving as a central point of access for users. This will facilitate a multi-faceted analysis of the effects of the transition by experts from various fields of expertise.

  The database will include the data from the laboratory measurements and the intercomparison



launches made at the sites. Furthermore, it will include coincident ancillary measurements from e.g. satellite overpasses and/or ground-based remote sensors such as those identified in Section 7. Making ancillary measurements available together with the radiosonde intercomparison data allows for in-depth analysis and understanding of the differences between the RS92 and RS41 radiosondes, and is commensurate with one of the key principles of GRUAN: to have measurement redundancy.

The data format of the files in the ancillary database will be CF-compliant NetCDF for ease of access and the database will be built for easy web-based data discovery and access. It will be available as it is being populated with data to enable scientific analysis from the outset.

Although it may not be possible to analyze all aspects of the data immediately, building a long-term database will enable exploitation by the expert community well into the future and represent 590 a substantial value-added outcome. The database availability will be advertised via the GRUAN website at https://www.gruan.org and readers should check this source for the latest status.

**8.4 Technical documentation**

Documentation is a foundation stone of a reference network such as GRUAN. It is essential for the transfer of knowledge, ranging from describing operational procedures and best practices to 595 perform measurements, via a detailed description of correction algorithms, to documenting changes to measurement systems. In a broader sense, robust documentation ensures the traceability of the data products, a requirement for reference data. Only through the existence of proper documentation is it possible to assure the quality of the measurement data within GRUAN. All GRUAN technical documentation is available on the GRUAN website under https://www.gruan.org/documentation/ 600 gruan/.

In the specific case of the RS92 to RS41 transition, comprehensive documentation will provide the required transparency on how the change was managed, and this will make it possible to reconstruct and scrutinize the reported differences between the RS92 and the RS41 radiosondes, even after many years. Furthermore, this documentation will serve as a template for managing any future 605 transitions of measurement systems within GRUAN. Finally, with GRUAN documentation available to the wider scientific community, other networks, such as GUAN or the Global Observing System (GOS) might be assisted in managing changes in their observational radiosonde systems.

This paper serves as an overarching document, outlining the strategy of managing the RS92-RS41 transition within GRUAN. Other GRUAN technical documents and publications will cover various 610 aspects in more detail. For example, a technical note was released which outlines the GRUAN recommendations for the rig configuration for performing twin soundings (von Rohden et al., 2016), whereas rig configurations for extended payloads consisting of multiple instruments and radiosondes are discussed in Jauhiainen et al. (2016). It is foreseen that separate papers will be written that report on:

– The results of the laboratory characterization of RS41 sensors described in Section 5





- – Synthesis of the RS92-RS41 intercomparison studies

- – Comparison against non-radiosonde measurements (e.g. ancillary data)

In addition, a final paper will be drafted that collates the results of the separate reports and summarizes and evaluates the outcomes of the RS92-RS41 transition for GRUAN.

**9 How to get involved**

The GRUAN change management programme envisaged in this paper could, in principal, be completed solely by current GRUAN members (sites, Lead Centre, scientists in the working group on GRUAN, and task teams). However, we explicitly recognize that there are substantial resources and expertise beyond the immediate GRUAN community which could increase the robustness of all as-
pects of the envisaged programme. Some specific potential suggestions are given below but there are undoubtedly many more ways to get involved.

Participation of non-GRUAN sites, who plan to undertake an intercomparison of their own, is strongly encouraged. Sites need not undertake the full multi-season campaign to contribute substantive value. Any additional intercomparison data will provide either additional training data sets or
a means to independently validate results and ensure that any geographical effects have been adequately accounted for. Sites should contact the Lead Centre staff (lead author) to initiate a discussion around data submission requirements.

Participation of experts in the analysis of the results is strongly encouraged. Research results are likely to be more robust and comprehensive after accounting for a broad range of user inputs.
The GRUAN community, although broadly diverse, likely misses some important types of expertise. The GRUAN Lead Centre and Working Group chairs can provide letters of support and further information to investigators wishing to apply for grant support to aid their involvement in the analysis of the transition from the RS92 to other models of radiosonde.

Dissemination and outreach of results leading to impact for both NRT and long-term applications
will require sustained community engagement. It is important that the research results translate to real-world applications and that, ultimately, will require user uptake.

**10 Summary and outlook**

In this paper we have described the ongoing GRUAN-wide coordinated approach to managing the change from the Vaisala RS92 to the RS41 as an operational radiosonde system within GRUAN.
Since the network's goal is to provide long-term reference-quality observations of ECVs such as temperature and water vapor for the purpose of e.g. climate monitoring, it is vital that this change of measurement system does not introduce discontinuities or inhomogeneities in the GRUAN data records. The majority of the 27 GRUAN sites were launching the RS92 as their operational ra-





diosonde, which was phased-out in the last quarter of 2017, and most of these sites have since
switched to the RS41.

Such a large-scale change in instrumentation is unprecedented in the history of GRUAN and
poses a challenge for the network. To ensure the integrity of the data record before and after the
transition, it is necessary to fully understand and characterize the differences, the bias adjustment,
and measurement uncertainty between the RS92 and RS41 radiosondes. Within GRUAN several
different, but aligned, programmes are generating a body of knowledge to underpin the RS92-RS41
transition, involving laboratory characterization of measurement errors due to external factors (e.g.
solar radiation, time-lag), extensive twin sounding studies with RS92 and RS41 on the same balloon,
and comparison with ancillary measurements.

Regular twin soundings for a period of two years is considered sufficient to capture seasonal-
ity in the differences. Since not all sites are able to implement such an extended intercomparison
programme, burden-sharing is employed, whereby at sites with similar climatological conditions,
only one site needs to perform the intercomparisons. All data relevant to the RS92-RS41 transition
are archived in a scientific database that will be accessible to the scientific community for exter-
nal scrutiny, enabling transparency and traceability. These data include radiosoundings, collocated
satellite observations and other ancillary measurements as well as the data from the laboratory mea-
surements. Furthermore, data from intercomparison studies performed at several non-GRUAN sites
have been shared with GRUAN by cooperating meteorological services and institutes and these are
also included in the database.

Preliminary analysis of the laboratory experiments indicates that the calibrations of the RS41's
temperature and humidity sensors are more accurate than for the RS92. Comparison with external
references show calibration uncertainties of $< 0.1 \, \text{K}$ for temperature and $< 1\% \, \text{RH}$ for the humidity
sensor. Preliminary analysis of 224 RS92-RS41 twin soundings performed at Lindenberg Observa-
tory show that differences between RS92-GDP.2 and Vaisala-processed RS41 nighttime temperature
measurements are smaller than $0.1 \, \text{K}$ over the entire profile. However, daytime temperature differ-
ences in the stratosphere increase steadily with altitude, with $T_{\text{RS92-GDP.2}}$ $0.6 \, \text{K}$ higher than $T_{\text{RS41}}$
at $35 \, \text{km}$. RH values measured by the RS41 in the troposphere are up to 8% higher. These higher
$\text{RH}_{\text{RS41}}$ values are consistent with the analysis of satellite-radiosonde collocations. A comprehensive
analysis of all twin soundings performed within GRUAN, that will also evaluate the effect of e.g.
climatological factors and will also include ancillary data, is ongoing.

Commensurate with the importance of detailed and comprehensive documentation to GRUAN's
operations, the RS92-RS41 transition will be extensively documented to ensure traceability of the
process. Furthermore, the documentation will help to convey the experience and knowledge gathered
to other networks to aid them in managing any changes in their operational radiosonde systems.
Future publications will report the results of the characterization of the RS92-RS41 differences from
various perspectives, and a final publication will evaluate the impact of the RS92-RS41 transition on
GRUAN data records.



*Acknowledgements.* The authors wish to thank David Smyth of Maynooth university, and other members of the
GRUAN community, for their contribution to the discussion of the GRUAN change management, which greatly
helped to improve this paper.



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





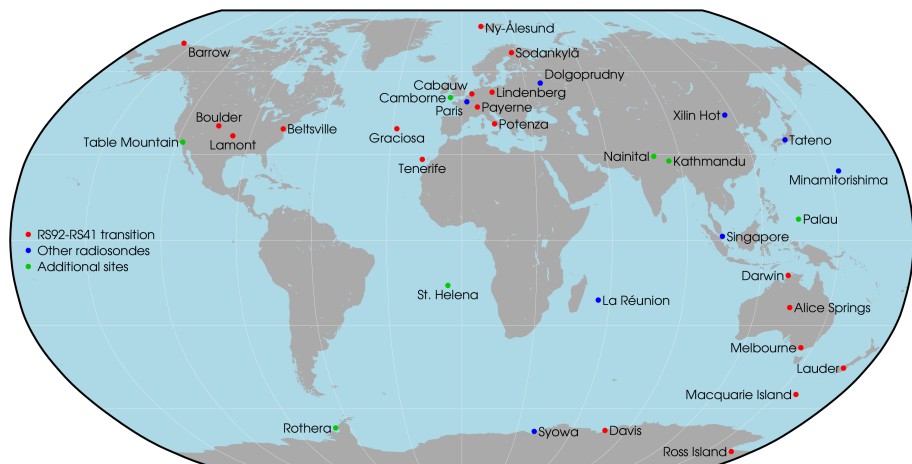

**Fig. 1.** Locations of GRUAN sites and other transition-related sites (September 2019). Red dots represent sites that switched from RS92 to RS41 as their operational radiosonde. Blue dots represent sites that employ another type of radiosonde or that transitioned from another type to the RS41. Green dots represent sites which performed additional comparison launches and provided these data to GRUAN.



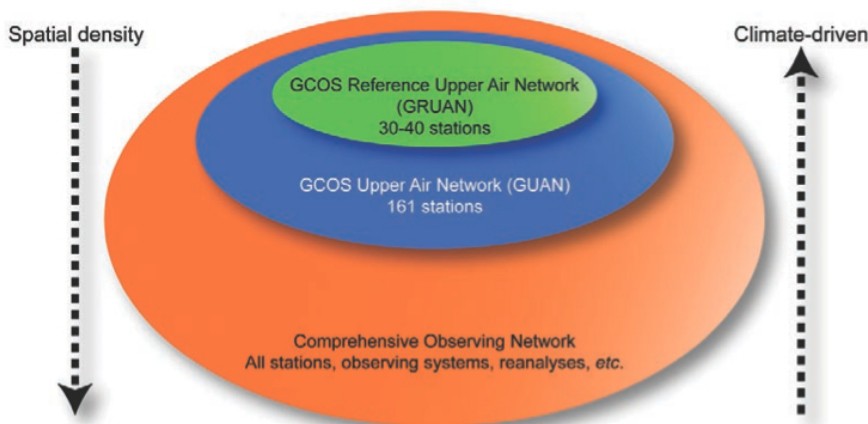

**Fig. 2.** Schematic diagram of the GCOS tiered-network concept of upper-air observations for climate. GRUAN offers less spatial density than either GUAN (a subset of the global radiosonde network stations that have made commitments of longevity and metadata collection) or the comprehensive network (which includes all ground-based and satellite upper-air observations and reanalyses). From Seidel et al. (2009).



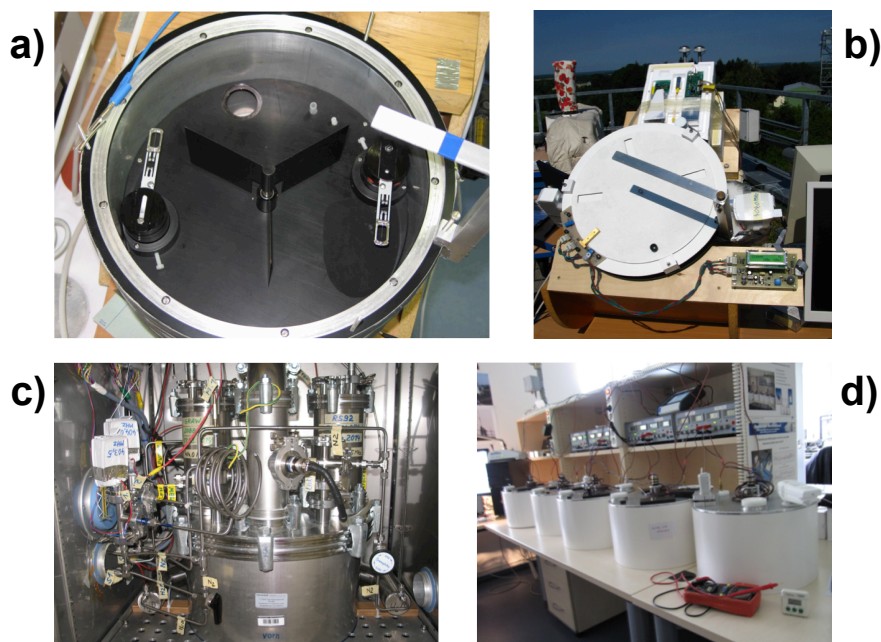

**Fig. 3.** Overview of the experimental equipment used to characterize the measurement errors and uncertainties of the RS92 and RS41 sensors. a) Radiation test chamber (top view) b) Radiation test chamber in operation, oriented perpendicular to the Sun (light source). Visible on top of the quartz plate is the shutter, with 3 opening slits, allowing for simultaneous testing of 3 radiosondes. c) Climate chamber that is used to test the performance of radiosondes under various temperature, pressure and humidity conditions. In the configuration shown here, the response time-lag of the humidity sensor is investigated. d) Standard humidity chambers that contain reference saline solutions that generate RH environments between 0 and 100 %RH.

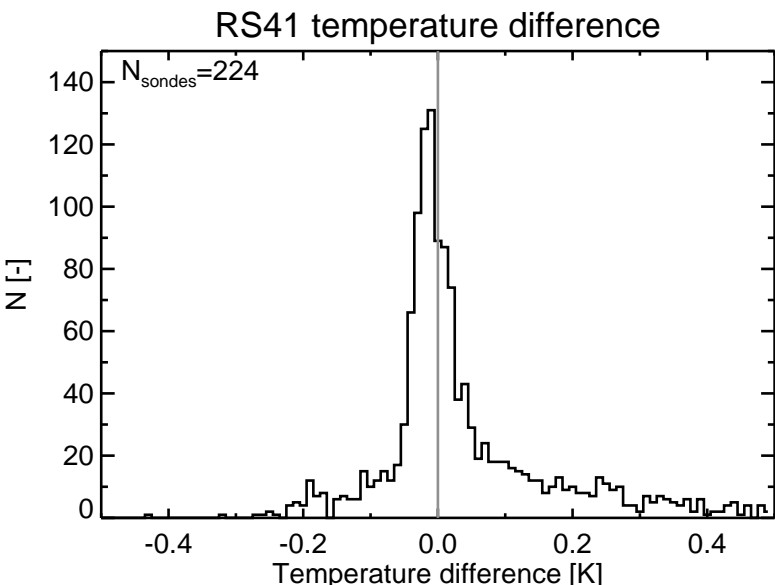

**Fig. 4.** Histogram of the differences between the temperature recorded by the RS41 and the reference Pt100 probe at room temperature under various humidity conditions inside a standard humidity chamber. The plot displays the collated results for tests in RH environments of 0, 11, 33, 75, and 100 %RH (Table 4) that were performed between 2014 and 2018. Data are collected in 0.01 K wide bins.



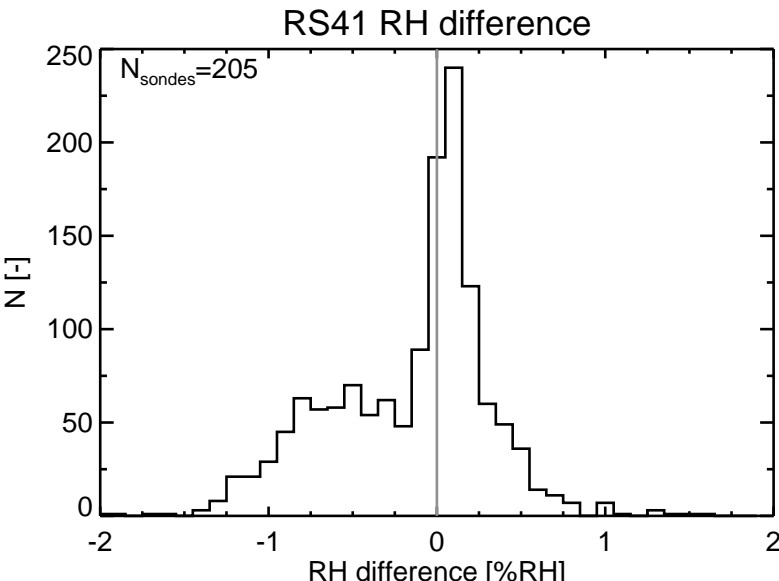

**Fig. 5.** Histogram of the differences between the RH value recorded by the RS41 and the reference value at room temperature for reference RH values of 0, 11, 33, 75, and 100 %RH in the standard humidity chamber (Table 4) that were performed between 2014 and 2018. The plot displays the collated differences for all 5 reference RH values. Data are collected in 0.1 %RH wide bins.



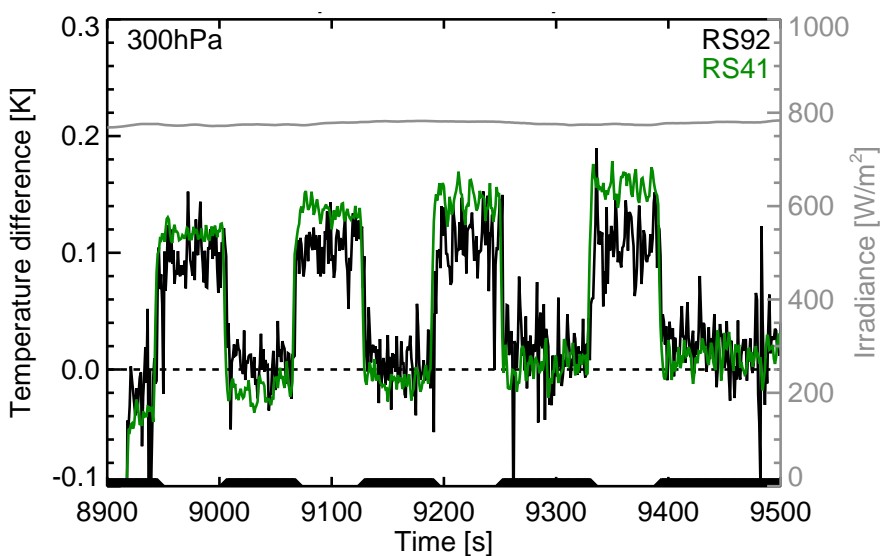

**Fig. 6.** Results of radiation tests with the RS41 (green) and RS92 (black) radioondes at $300\,\mathrm{hPa}$ with 5 m/s ventilation flow. The solar irradiance during the test (thin grey trace) was approximately $800\,\mathrm{W/m^2}$. The thick black segments at zero temperature difference indicate when the shutter was closed. During shutter open periods the sensor was illuminated for 60 s.

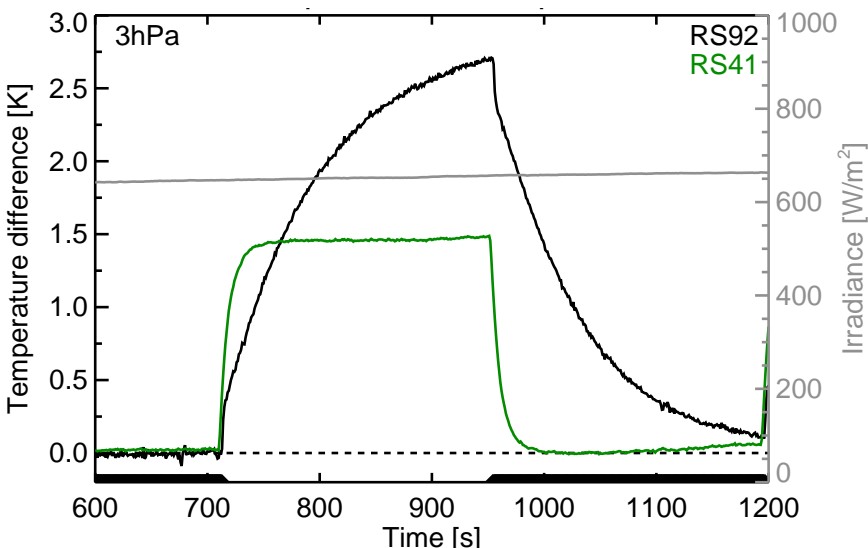

**Fig. 7.** Results of radiation tests with the RS41 (green) and RS92 (black) radiosondes at $3\,\mathrm{hPa}$ with $5\,\mathrm{m/s}$ ventilation flow. The solar irradiance during the test (thin grey trace) was approximately $650\,\mathrm{W/m^2}$. The thick black segments along the x-axis indicate when the shutter was closed. During shutter open periods the sensor was illuminated for $240\,\mathrm{s}$.



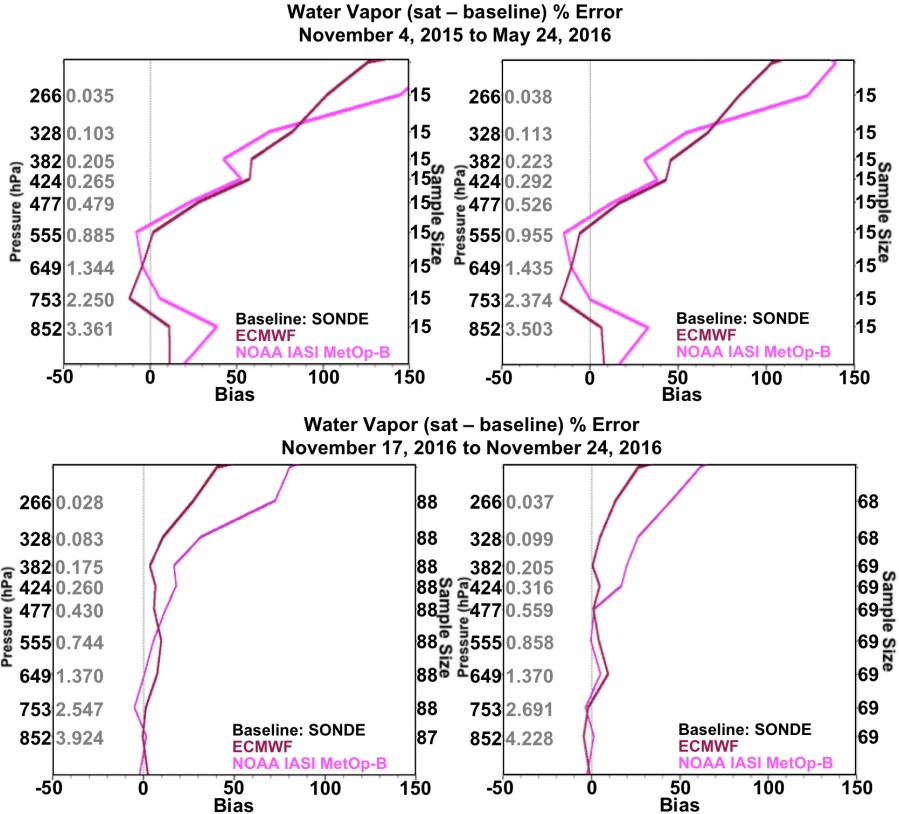

**Fig. 8.** Mean biases between water vapor profiles from radiosonde and satellite observations from IASI MetOp-B (red). The relative differences are expressed as $100 \times (\text{WVMR}_{sat} - \text{WVMR}_{radiosonde})/\text{WVMR}_{radiosonde}$. The differences with associated ECMWF analysis data are represented by the purple trace. The relative differences with the RS92 are shown in the left-hand column, differences with RS41 in the right-hand column. The upper panels present the results for the GRUAN site in Lauder, New Zealand, whereas the lower panels present the results for radiosoundings performed during a 10-day period at conventional radiosonde sites in Europe ($35$–$65°$N, $10°$W $-40°$E). Coincidence criteria: less than 1 hour (2 hours for ECMWF data) at approximately $500\,\text{hPa}$ and within $50\,\text{km}$ at the surface. The labels at the y-axis depict the pressure level (black); the average water vapor mixing ratio (g/kg) is given in grey. The numbers at the right-hand y-axis represent the number of coincident observations at each pressure level.



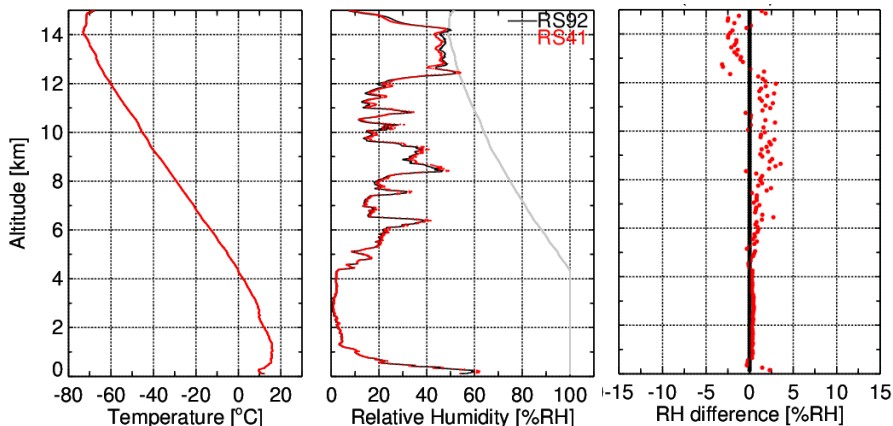

**Fig. 9.** Twin sounding with RS92 and RS41 on the same balloon performed on 2 November 2015, 12 UTC, at Lindenberg meteorological Observatory. Left-hand panel shows the observed temperature profile of RS92 (black) and RS41 (red). The middle panel shows the observed relative humidity profile over water, the grey trace represents ice saturation. The right-hand panel shows the differences between the humidity of both sondes (RH$_{41}$ - RH$_{92}$), the red dots represent data collected in 500 m wide altitude bins.





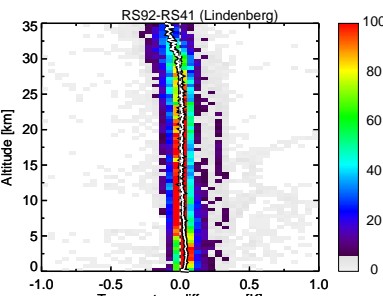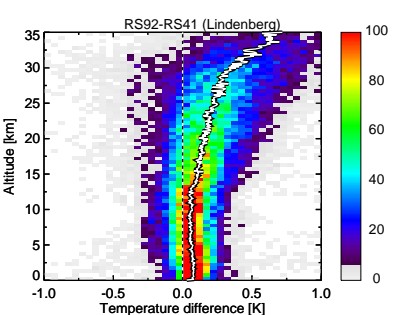

**Fig. 10.** Temperature differences between RS92-GRUAN data product (RS92-GDP.2) and Vaisala-processed RS41 data (RS92$_{GDP.2}$–RS41$_{EDT}$) for comparion soundings performed at the Lindenberg GRUAN site. For each sounding profile the mean $T_{RS92\text{-}GDP.2}$-$T_{RS41}$ was calculated in a 500 m-wide altitude bin. Data are presented as a scatter density plot in altitude-temperature bins of 0.5 km x 0.05 K. The color represents the number of data points in each bin. The thick white trace represents the averages of the temperature differences for each 500 m-wide altitude bin. Left-hand panel displays the results for 77 nighttime soundings, whereas the right-hand panel shows the results for 147 daytime soundings.

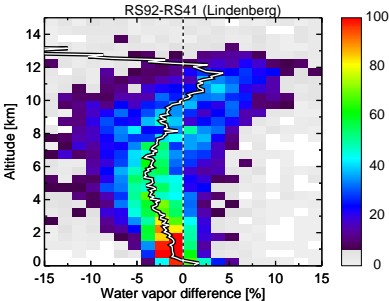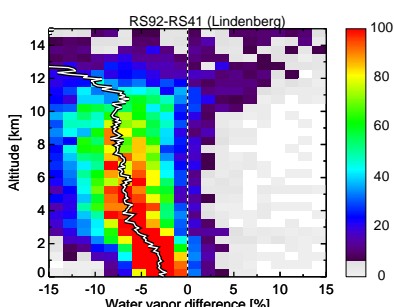

**Fig. 11.** Same as Figure 10 but for relative humidity. Shown is the relative difference 100x(RH$_{RS92\text{-}GDP.2}$–RH$_{RS41}$)/RH$_{RS92\text{-}GDP.2}$. Bin-size for the scatter density plot is 0.5 km x 2.5% RH. The average tropopause height over Lindenberg is approximately 10-12 km.





Table 1: List of GRUAN sites that are involved in the RS92-RS41 transition. Listed are: the dates at which a site switched from RS92 to RS41 as operational radiosonde for the sites, the length and frequency of the twin sounding programs, and the number of twin soundings performed (y - year, m - month, w - week, d - day, h - hour).

| Site | Position (Lat °N/Lon °E) | Change Date | Duration, Interval | Amount (+ ongo.) | Remark |
|---|---|---|---|---|---|
| Alice Springs | –23.79/133.89 | 2018-08-22 | – | – | – |
| Barrow | 71.32/–156.61 | 2017-12-15 | – | – | ARM RIVAL campaign canceled |
| Beltsville | 39.05/–76.88 | – | 3 y, ≥1 w | 126+ | regular twin soundings (2016-12 to 2019-09, ongoing), no daily operational soundings |
| Boulder | 39.95/–105.20 | 2017-01 | – | – | no daily operational soundings |
| Cabauw | 51.97/4.92 | 2017-05-11 | – | – | – |
| Darwin | –12.43/130.89 | 2018-07-01 | 1 m, 12 h | 57 | campaign (2018-06) |
| Davis | –68.57/77.97 | 2019-02-23 | – | – | – |
| Graciosa | 39.09/–28.03 | 2019-01-12 | 1 y, ≥1 w | 30 | ARM RIVAL campaign (2018-04 to 2019-03) |
| La Reunion | –21.08/55.38 | – | 1 w + 1 w, 1 d | 38 | two short campaigns (2015-05 and 2019-01), no operational use of RS92 and RS41 |
| Lamont | 36.6/–97.49 | 2017-12-15 | 1 w, ≤5/d | 20 | one short campaign (2014-06) (see Jensen et al., 2016). |
| | | | 1 y, ≥1 w | 49 | ARM RIVAL campaign (2018-02 to 2019-04) |
| Lauder | –45.05/169.68 | 2016-07, 2018-03-28 | 1 y, 1 w | 57 | regular twin soundings (2015-10 to 2016-11) |
| Lindenberg | 52.21/14.12 | 2017-03-22 | 5 y, 1 w to 12 h | 410+ | intensive campaigns in addition to weekly twin soundings (2014-12 to 2019-07, ongoing) |
| Macquarie Island | –54.50/158.94 | 2019 | – | – | transition planned for 2019 |
| Melbourne | –37.67/144.83 | 2018-04-01 | – | – | – |

*This table is continued on the next page.*





| Site | Position (Lat °N/Lon °E) | Change Date | Duration, Interval | Amount (+ ongo.) | Remark |
|---|---|---|---|---|---|
| Ny-Alesund | 78.92/11.93 | 2017-04-01 | 3 y, 1 w | 125 | regular twin soundings (2015-03 to 2018-03) |
| Payerne | 46.81/6.95 | 2018-03-29 | 4.5 y, ∼1 w | 143 | regular twin soundings (2014-08 to 2018-12), switch of operational radiosonde: Meteolabor SRS-C50 to RS41 |
| Potenza | 40.60/15.72 | 2016-11 | – | – | no daily operational soundings |
| Ross Island | –77.85/166.66 | 2016-02-12 | – | – | – |
| Sodankyla | 67.37/26.63 | 2017-03-28 | – | 8 | sporadic twin soundings (2015-10 to 2018-07) |
| Tateno | 36.06/140.13 | – | 1 w, 12 h | 10 | campaign, switch of operational radiosonde: RS92 to Meisei RS-11G |
| Tenerife | 28.32/–16.38 | 2017-12-13 | – | – | – |



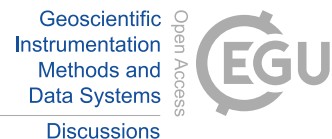

**Table 2.** Non-GRUAN sites that provided data of RS92-RS41 intercomparison flights to GRUAN. (y - year, m - month, w - week, d - day, h - hour).

| Site | Position (Lat °N/Lon °E) | Duration, Interval | Amount | Remark |
|---|---|---|---|---|
| Camborne | 50.22/–5.33 | 2 w, ≤4/d | 120 | campaign organized by Vaisala in collaboration with UK MetOffice (2013-11), 30 flights with 2xRS92 and 2xRS41 on one rig (see Edwards et al., 2014) |
|  |  | 7 m, ≥1 w | 25 | regular twin soundings (2018-04 to 2018-10) |
| Kathmandu | 27.62/85.54 | 2 w, ≤4/d | 28 | StratoClim campaign (2017-08) |
| Nainital | 29.35/79.46 | 1 m + 1 w, ≤3/d | 35 | StratoClim campaign (2016-08, 2016-11) |
| Palau | 7.34/134.47 | 3 w, 48 h | 20 | StratoClim campaign (2018-03, 2018-10, 2019-05) |
| Rothera | –67.57/–68.13 | 1 y, ≥1 w | 48 | regular twin soundings (2017-05 to 2018-08) |
| St. Helena | –15.94/–5.67 | 1 y, ≥1 w | 51 | regular twin soundings (2017-12 to 2018-12) |
| Table Mountain | 34.39/–117.7 | 5 y, sporadic | 53 | sporadic twin soundings (2014 to 2018) |
| Ship R/V Mirai | (cruise) | 5 m, sporadic | 36 | sporadic twin soundings (2015-08 to 2015-12), (see Kawai et al., 2017) |





**Table 3.** Overview of laboratory experiments performed at the GRUAN Lead Centre to characterize the RS41 sensors.

| Experiment | Sensor | Range | Set-up | Time frame |
|------------|--------|-------|--------|------------|
| Radiation | T, RH | 3-1013 hPa | radiation set up | 2014-2020 |
| Time-lag | RH | –75°C - –40°C | climate chamber | 2015-2020 |
| Calibration | RH | 0-100 % | SHC, reference salts | since 2014 |





**Table 4.** Humidity values achieved in the SHC using desiccant, pure water and 4 different saturated salt solutions.

| Relative Humidity (%RH) | 0 | 11 | 33 | 75 | 97 | 100 |
|---|---|---|---|---|---|---|
| SHC contents | desiccant | LiCl | MgCl | NaCl | $K_2SO_4$ | distilled $H_2O$ |





**Table 5.** List of scientific analysis tasks with their corresponding lead investigator(s).

| Task | Lead |
| --- | --- |
| Laboratory experiments | |
| Radiation tests | Lead Centre |
| Time-lag | Lead Centre |
| Development GRUAN data product for RS41 | Lead Centre |
| Twin soundings | Lead Centre + Task Team (TT) radiosonde |
| Ancillary measurements | TT ancillary measurements |
| NPROVS | Tony Reale |
| Metrology | Andrea Merlone |
| Statistical analysis | Alessandro Fasso |
| Evaluation of RS92-RS41 transition | Lead Centre/Working Group |