# Peer review of "Progress in managing the transition from the RS92 to the Vaisala RS41 as the operational radiosonde within the GCOS Reference Upper-Air Network"

_Geoscientific Instrumentation, Methods and Data Systems, 2019_

## Referee Comment (RC1) · Anonymous Referee #1 · 29 Dec 2019

This manuscript reports about estimation methods and estimates of differences between Vaisala RS92 and RS41 radiosondes. It is an interim report, not a final report, but yet it already contains a wealth of valuable information on what to expect when switching from RS92 to RS41 radiosondes. As can be expected by a reference network like GRUAN, utmost care is taken to characterize measurement errors through laboratory experiments, twin launches and comparison with anciliary satellite information. Results of the comparisons are already quite robust, however some comparisons particularly in the Tropics and over Antarctica have yet to be completed. The authors

ask for active participation of radiosonde station operators and also offer logistical help so that the number of twin launches can be maximized. The authors also describe in some detail how the information about the transition will be provided to interested users. From the experience with already available sounding data from GRUAN, one can expect a mature presentation of those results.

I have very little to criticize and rather would suggest to publish this manuscript quickly.

There are only two comments from my side:

The title is rather long and likely a better title can be found. It does not need to be so specific. Suggestions from my side are: "Managing the transition from Vaisala RS92 to RS41 radiosondes within the GRUAN: a progress report" or "Fully traceable switch from Vaisala RS92 to RS41 radiosondes within the GRUAN: a progress report". I think one should either write GRUAN or "Global Climate Observing System Reference Upper-Air Network", since there will be few readers who know GCOS but do not know GRUAN.

There is some redundancy in stating the importance of characterizing the errors and differences. For example, the description of differences with the transition handling at Tateno (l130-136) could be omitted.

---

## Referee Comment (RC2) · Anonymous Referee #2 · 21 Feb 2020

**REFEREE'S REPORT**

Manuscript number:     Geosci. Instrum. Method. Data Syst.; https://doi.org/10.5194/gi-2019-36
Title:     Progress in managing the transition from the RS92 to the Vaisala RS41
as the operational radiosonde within the GCOS Reference Upper-Air
Network

**Abstract**
- Lines 22 - 24: See my comments on section 5.2: corresponding modifications are needed here.

**1 Introduction**
- Lines 32 - 35: The sentence "The network consists … in the processing chain." is very long, complicated and difficult to follow for a reader.
- Line 36: "the only GRUAN data products"
  - Should this be " the only sources of the GRUAN data products"?

**3 Change management**
- Line 192: "each other's uncertainty coverage factor"
  - the coverage factor only specifies the confidence level at which an uncertainty is given; therefore this text should probably be "the uncertainties of the data at the 95 % confidence level"
- Lines 207 - 208: "consider the metrological quantification aspects that are necessary in the current case."
  - such as? to which aspects are referred here?

**5.2 Results of the laboratory characterization**
- There is a clear shortcoming in Section 5: This paper emphasizes several times the metrological aspects but no uncertainty values are given for the calibration systems in Section 5.
- Line 324: "from its width it can be inferred that the calibration uncertainty is smaller than 0.1 K."
  - Such conclusion cannot be drawn without knowing the uncertainty of the reference. (Basically, you may possibly conclude that the equivalence between your reference and Vaisala's reference is within 0.1 K but the link to the SI realisations is still missing.)
- Lines 328 - 329: "which means that the uncertainty of the humidity calibration is smaller than 1 %RH."
  - Same comment as for temperature above.
  - Moreover, the uncertainty of the reference (both at Vaisala and Lead Centre) is significantly at different RH-levels; therefore analysing them together in single histogram is missleading.
  - Asymmetry of the histogram indicates probably some systematic differences between Vaisala and Lead Centre at some measurement points.
  - Finally, for me it looks strange that in Fig. 4 all visible data fall in claimed uncertainty but here a significant amount of data fall out from the range of claimed uncertainty.

**6 Metrology**
- Lines 400 - 414:
    o It's unclear to me why these two last paragraphs of this Section are needed here: I suggest focusing them in facilities/services/results developed/obtained at national metrology institutes that can be benefited in the transition from RS92 to RS41 (and in radiosonde measurements in general)

**10 Summary and outlook**
- Lines 670 - 672 "2. Comparison with external references show calibration uncertainties of < 0.1 K for temperature and < 1% RH for the humidity sensor."
    o Ref. to my comments in Section 5.2, I don't agree with this sentence.

---

## Author Response (AR1)

[Review comments in red, author response in black]

Reviewer #1

The title is rather long and likely a better title can be found. It does not need to be so specific. Suggestions from my side are: "Managing the transition from Vaisala RS92 to RS41 radiosondes within the GRUAN: a progress report" or "Fully traceable switch from Vaisala RS92 to RS41 radiosondes within the GRUAN: a progress report". I think one should either write GRUAN or "Global Climate Observing System Reference Upper-Air Network", since there will be few readers who know GCOS but do not know GRUAN.

Thank you for the suggestion. We will change the title to
"Managing the transition from Vaisala RS92 to RS41 radiosondes within the Global Climate Observing System Reference Upper-Air Network (GRUAN): a progress report"

There is some redundancy in stating the importance of characterizing the errors and differences. For example, the description of differences with the transition handling at Tateno (l130-136) could be omitted.

We would prefer to keep the lines 130-136 because they summarize the differences in the scale of the approaches between:
a) an instrument change at a single site and in contrast,
b) the GRUAN approach to maintain the consistency of the data records at multiple sites within a network, which includes a coordinated network-wide effort, an extensive laboratory characterization, and the call on expertise from various fields that are represented in the network.

[Review comments in red, author response in black]

Reviewer #2

**Abstract**
Lines 22 – 24: See my comments on section 5.2: corresponding modifications are needed here.
Agreed. The text will be changed to read
"[....] with uncertainties of < 0.2 K for the temperature and < 1.5% RH for the humidity sensor."

**1 Introduction**
Lines 32 – 35: The sentence "The network consists … in the processing chain." Is very long, complicated and difficult to follow for a reader.
The sentence will be rephrased and restructured to increase readability.

Line 36: "the only GRUAN data products" Should this be " the only sources of the GRUAN data products"?
For clarity the sentence will be changed to start with
"[...]the GRUAN data products for the Vaisala RS92 and the Meisei RS-11G radiosondes are the only [...]"

**3 Change management**
Line 192: "each other's uncertainty coverage factor"
the coverage factor only specifies the confidence level at which an uncertainty is given; therefore this text should probably be "the uncertainties of the data at the 95% confidence level"
We agree with this comment and the sentence will be changed accordingly.

Lines 207 - 208: "consider the metrological quantification aspects that are necessary in the current case."
such as? to which aspects are referred here?
Aspects such as type B (instrumental) expanded uncertainties (absolute for each radiosonde), relative uncertainty in the comparison, traceability of involved instrumentation, and assessment of the comparability with independent ancillary results.
These are referred to here since this is the key aspect in adopting a metrological approach, where other reference instruments are adopted to make the evaluation traceable to standards and with a documented uncertainty budget. The manuscript will be modified to include these aspects listed above.

**5.2 Results of the laboratory characterization**
There is a clear shortcoming in Section 5: This paper emphasizes several times the metrological aspects but no uncertainty values are given for the calibration systems in Section 5.
Thank you for pointing out this omission. We will include information about the uncertainty of the references for the temperature and relative humidity.

The uncertainty of the reference Pt100 temperature sensor is 0.04K (1 sigma), certified by DAkkS (the national accreditation agency in Germany).
The uncertainty of the relative humidity of the air over the reference salt mixtures is at most 0.5%RH, as reported by Greenspan (1977) doi: 10.6028/jres.081A.011.
A reference to Greenspan (1977) and a statement about the calibration uncertainty of the reference temperature sensor will both be included in the updated manuscript.

Line 324: "from its width it can be inferred that the calibration uncertainty is smaller than 0.1 K."
o Such conclusion cannot be drawn without knowing the uncertainty of the reference. (Basically, you may possibly conclude that the equivalence between your reference and Vaisala's reference is within 0.1 K but the link to the SI realisations is still missing.)
A link to SI realisations is provided by the certification by DAkkS. The calibration uncertainty of the Pt100 0.04K (1 sigma).
For clarity, Figure 4 will be updated to show the percentiles that mark the boundaries of 1-sigma and 2-sigma ranges. The boundaries show that the calibration uncertainty of the RS41s temperature sensor is within 0.2K (1 sigma).
The text will be changed to reflect this.

Lines 328 - 329: "which means that the uncertainty of the humidity calibration is smaller than 1 %RH."
o Same comment as for temperature above.
The uncertainty of the relative humidity of the air over the reference salt mixture at room temperature is at most 0.5%RH, as reported by Greenspan (1977).
Table 4 will be updated to include the uncertainty in the relative humidity of the air over each reference salt mixture.
Taking into account the uncertainty of the RH references and the histogram distributions at each tested RH level (discussed further in the following points), we will change this statement to "[..]the uncertainty of the humidity calibration is smaller than 1.5 %RH."

o Moreover, the uncertainty of the reference (both at Vaisala and Lead Centre) is significantly at different RH-levels; therefore analysing them together in single histogram is misleading.
In Figure 5, histograms will be added for the measurements at 0, 11, 33, 75, 100 %RH. We will discuss the calibration uncertainty of the radiosonde's RH sensor at each of these levels.

o Asymmetry of the histogram indicates probably some systematic differences between Vaisala and Lead Centre at some measurement points.
The asymmetry of the histogram in Figure 5 is due to an error in the analysis, where we used incorrect reference values at 33%RH. We want to thank the reviewer for this comment, which prompted us to re-check our analysis of the measurements. The incorrect reference values at 33%RH caused a shoulder in the histogram at around -0.75%RH. With the correct reference values, this

shoulder disappears and the histogram distribution becomes more symmetric, although a slight asymmetry remains.

This remaining asymmetry is caused by the positions and shape of the histograms at different RH levels, which does indeed show discrepancies between the reference salt mixtures and the calibration by Vaisala. A statement to this effect will be added in the manuscript.

o Finally, for me it looks strange that in Fig. 4 all visible data fall in claimed uncertainty but here a significant amount of data fall out from the range of claimed uncertainty.

The updated Figure 4 shows that ±0.5 K covers more than 2 sigma of the distribution of the differences between the RS41 temperature sensor and the Pt100 reference. In fact only 32 of 1406 measurements (2.2%) are outside the ±0.5 K range.

We agree that due to the non-Gaussian shape of the distribution ±0.1K does not cover 1 sigma, and that ±0.2K is more appropriate. The text will be changed accordingly.

**6 Metrology**
Lines 400 - 414:
o It's unclear to me why these two last paragraphs of this Section are needed here: I suggest focusing them in facilities/services/results developed/obtained at national metrology institutes that can be benefited in the transition from RS92 to RS41 (and in radiosonde measurements in general)

The last two paragraphs will be removed, but we would like to keep a sentence referring to the Meteomet project, that concerns the investigation of environmental factors on the uncertainty of meteorological measurements and therefore is relevant to the paper.

**10 Summary and outlook**
Lines 670 - 672 "2. Comparison with external references show calibration uncertainties of < 0.1 K for temperature and < 1% RH for the humidity sensor."
o Ref. to my comments in Section 5.2, I don't agree with this sentence.

Agreed. The text will changed to read uncertainties of < 0.2 K for temperature and < 1.5% RH

[Figure]

Updated Figure 4

[Figure]

Updated Figure 5.
[internal comment for the co-authors] In the previous version of the plot, the
data for 33%RH were shifted towards -75%RH due to an error in the analysis.

This has now been corrected. The peaks of the histograms at the various RH levels don't coincide, which is due to differences between the Vaisala calibration and the reference salts. The reference salts are pretty good (uncertainty ~0.2%RH) so I think the differences at 75 and 100%RH are due to Vaisala's calibration. This RH-dependent error is consistent with SHC tests we performed with the RS92.

Manuscript prepared for Geosci. Instrum. Method. Data Syst.
with version 5.0 of the LATEX class copernicus.cls.
Date: 14 May 2020

**Progress in managing the transition from the RS92 to the Vaisala RS41 as the operational radiosonde within the GCOS Reference Upper-Air Network**

Ruud J. Dirksen[1], Greg E. Bodeker[2], Peter W. Thorne[3], Andrea Merlone[4], Tony Reale[5], Junhong Wang[6], Dale F. Hurst[7,8], Belay B. Demoz[9], Tom D. Gardiner[10], Bruce Ingleby[11], Michael Sommer[1], Christoph von Rohden[1], and Thierry Leblanc[12]

[1]GRUAN Lead Centre, Deutscher Wetterdienst, Meteorologisches Observatorium Lindenberg, Am Observatorium 12, 15848 Tauche/Lindenberg, Germany
[2]Bodeker Scientific, Alexandra, New Zealand
[3]Maynooth University, Maynooth, Ireland
[4]INRI, Turin, Italy
[5]NOAA/NESDIS, Washington DC, USA
[6]Department of Atmospheric and Environmental Sciences, State University of New York, Albany, USA
[7]Global Monitoring Laboratory, NOAA Earth System Research Laboratories, Boulder, CO, USA
[8]Cooperative Institute for Research in Environmental Sciences, University of Colorado, CO, USA
[9]Howard University, Washington D.C., USA
[10]National Physical Laboratory, Teddington, UK
[11]ECMWF, Reading, UK
[12]JPL, Pasadena, USA

Correspondence to: R. J. Dirksen (Ruud.Dirksen@dwd.de)

**Abstract.** This paper describes the GRUAN-wide approach to manage the transition from the Vaisala RS92 to the Vaisala RS41 as the operational radiosonde. The goal of the GCOS Reference Upper-Air Network (GRUAN) is to provide long-term high-quality reference observations of upper air Essential Climate Variables (ECVs) such as temperature and water vapor. With GRUAN data being used for climate monitoring, it is vital that the change of measurement system does not introduce inhomogeneities in to the data record. The majority of the 27 GRUAN sites were launching the RS92 as their operational radiosonde, and following the end of production of the RS92 in the last quarter of 2017, most of these sites have now switched to the RS41. Such a large-scale change in instrumentation is unprecedented in the history of GRUAN and poses a challenge for the network. Several measurement programmes have been initiated to characterize differences in biases, uncertainties and noise between the two radiosonde types. These include laboratory characterization of measurement errors, extensive twin sounding studies with RS92 and RS41 on the same balloon, and comparison with ancillary data. This integrated approach is commensurate with the GRUAN principles of traceability and deliberate redundancy. A two-year period of regular twin soundings is recommended, and for sites that are not able to implement this burden sharing is employed, such that measurements at a certain site are considered representative of other sites with similar climatological characteristics. All data relevant to the RS92-RS41 transition are archived in a database that

will be accessible to the scientific community for external scrutiny. Furthermore, the knowledge and experience gained about GRUAN's RS92-RS41 transition will be extensively documented to ensure

20  traceability of the process. This documentation will benefit other networks in managing changes in their operational radiosonde systems.

Preliminary analysis of the laboratory experiments indicates that the manufacturer's calibration of the RS41's temperature and humidity sensors is more accurate than for the RS92; with uncertainties of $<$ 0.2 K for the temperature and $<$ 5 %RH for the humidity sensor. A first analysis

25  of 224 RS92-RS41 twin soundings at Lindenberg Observatory shownighttime temperature differences $<$0.1 K between the Vaisala-processed temperature data of the RS41 ($T_{RS41}$) and the GRUAN dataproduct for the RS92 ($T_{RS92\text{-}GDP.2}$). However, daytime temperature differences in the stratosphere increase steadily with altitude, with $T_{RS92\text{-}GDP.2}$ up to 0.6 K higher than $T_{RS41}$ at 35 km. $RH_{RS41}$ values are up to 8 % higher, which is consistent with the analysis of satellite-radiosonde collocations.

**1 Introduction**

The Global Climate Observing System (GCOS) has instigated the GCOS Reference Upper-Air Network (GRUAN) (Bodeker et al., 2016) to perform reference quality measurements of upper-air Essential Climate Variables (ECVs) (Bojinski et al., 2014).

35   The network consists of a range of national contributions of high-quality observing facilities that undertake observations in a systematically similar manner. The employed method of observation and subsequent data processing 
[revised manuscript text omitted]

$$|m_1 - m_2| < k \cdot \sqrt{u_1^2 + u_2^2} \tag{1}$$

Verifying this consistency requires a sufficient population of coincident measurements to determine  that the data satisfy this condition in a statistically robust manner. Neither the GRUAN Manual (WMO, 2013a) nor the GRUAN Guide (WMO, 2013b) provide a clear requirement for the duration and intensity (measurement frequency) of an intercomparison study because it is inherently dependent on instrument type and measurement principle. However, as mentioned in Section 1, the manual on the GOS suggests a one-year intercomparison period. Different instruments measure different ECVs in very distinct manners. Some are episodic remote sensing, some are continuous remote sensing, others are periodic  in situ profiles. For each measurement type a different 'side-by-side' operation strategy would be required, which depends on (at a minimum): the instrument characteristics, variability in the target measurand, and cost and logistical considerations. Kobayashi et al. (2012) suggest that a total of 120 twin soundings spread across the seasonal cycle, at a given location, would be more than sufficient to characterize the effects of a change in radiosonde instrumentation, although the study did not consider the metrological quantification aspects that are necessary in the current case. Hence, for radiosondes, the GRUAN Lead Centre recommended that sites perform weekly, or with a  two-week interval, twin soundings for a period of two years. The two-year period ensures that seasonality is better probed, and a sounding interval of a week instead of a day mitigates the additional operational costs. The twin soundings should be equally distributed between day and night soundings. Above-mentioned metrological aspects refer to factors such as type B (instrumental) expanded uncertainties (absolute for each radiosonde), relative uncertainty in the comparison, traceability of involved instrumentation, and assessment of the comparability with independent ancillary results. This is the key aspect in adopting a metrological approach, where other reference instruments are used to make the evaluation traceable to standards and with a documented uncertainty budget.

[revised manuscript text omitted]
 different corrections for radiative heating are applied to the two sonde models by the Vaisala processing software.

The calibration accuracy and hysteresis effects of the humidity sensor have been investigated by placing the radiosonde's sensor boom in an SHC  with a stable, well-defined RH between 0 % and 100 %. The stable RH environment inside each SHC is achieved using one of the saline mixtures listed in Table 3. In addition, each SHC is equipped with a Pt100 reference thermometer which tests the calibration accuracy of the radiosonde temperature sensor. The reference Pt100 temperature sensors were certified by DAkkS (the German national accreditation agency), showing an uncertainty of 0.04 K ($1\sigma$). The RH values over reference salt mixtures are taken from Greenspan (1977), who specifies the upper limit of the uncertainty of the employed RH values at 25°C as 0.5 %RH.

The response time-lag of the humidity sensor is determined by measuring its reaction to a stepwise change in the humidity of the airflow, while keeping the temperature of the airflow constant. The time-lag effect becomes significant at temperatures below –40°C, the point at which the response time of the humidity sensor starts to exceed 10 s (Miloshevich et al., 2004; Dirksen et al., 2014). The time-lag tests are performed in a climate chamber which can reach temperatures as low as –75°C. Table 4 summarizes the laboratory experiments that have been performed to date to characterize the RS41 radiosonde.

The lab-based characterization results will be included in the scientific database holding all data

that are relevant to the RS92 to RS41 transition (discussed in Section 8.3).

**5.2 Results of the laboratory characterization**

To assess the calibration accuracy of the humidity and temperature sensors, more than 150 RS41
radiosondes from various production batches were tested in SHCs at the relative humidities listed
in Table 3. In a typical experiment, each RS41 was sequentially placed in a series of six SHCs
with increasing relative humidity, from 0 to 100 %, then back through the sequence of drier SHCs
to 0 %RH. This sequence also allows assessment of hysteresis of the humidity sensor. At each
humidity level, the radiosonde's sensor boom was inserted in the SHC for
approximately 4 minutes while its readings were recorded. The air inside each SHC was circulated
at 5 m/s by a fan, and the temperatures of the air and saline solution inside the SHCs were measured
by Pt100 reference thermometers. For some salts both air and solution temperatures are needed to
accurately determine the relative humidity in the SHC, since the humidity over the reference salt
depends on both quantities.

Figure 4 shows that the majority of the temperature measurements by the RS41 are within $\pm 0.5$ K
of the reference temperature. Although the tail of the distribution extends to 1 K (not shown), ~~only
a~~fewer than 2 % of the measurements show differences beyond 0.5 K. The mode of the distribution
indicates a bias of -0.025 K, while ~~from its width it can be inferred that the calibration uncertainty
is smaller than 0.1 K.~~a calibration uncertainty of $<0.2$ K $(1\sigma)$ can be inferred from the sensor's
uncertainty $\pm 0.04$ K and the 16th and 84th percentiles of the differences. As is apparent from
the percentile marks, the histogram of temperature differences between the RS41 and the Pt100
reference is not Gaussian, something that cannot yet be explained but will be the subject of further
investigation.

Figure 5 shows that all humidity measurements by the RS41 are within 2 %RH of the SHC refer-
ence RH value , and only a small fraction of the measurements showdifferences larger than
1 %RH . The
uncertainty of the humidity calibration is $<1.5$ %RH given the $<0.5$ %RH uncertainties of the SHC
RH references listed in Table 3.

~~The histogram of RH differences between the RS41 and the reference RH value has a pronounced
peak around 0 %RH, due to the fact that the calibration errors are dependent on RH. At low RH
the differences, and their spread, between the reference and the RS41 values are small, whereas
at high RH the differences and spread are larger. As a result, all observed calibration errors at
0 %RH humidity cluster around 0 %, thereby over-representing these values in the distribution.~~ The
histograms of RH differences at the various RH levels, represented by the colored traces in Fig 5,
show that the calibration errors are dependent on RH. At low RH the differences and their spread are
small, whereas at high RH the differences and spread are larger. As a result, the observed calibration
errors at 0 %RH humidity cluster around 0 %, (blue trace in Fig 5), thereby over-representing these

values in the overall distribution (black trace in Fig 5). The larger differences and spread at high RH indicate that there are discrepancies between the reference salt mixtures and the calibration by Vaisala.

Radiation tests were conducted in the modified SHC as described by Dirksen et al. (2014). Measurements were performed at various settings within the ranges:

– pressure between 3 hPa and ambient,

– irradiance from 200 to 1000 W/m$^2$,

– ventilation speed of either 2.5 or 5 m/s,

– illumination times of 1-4 minutes.

The illumination times depended on pressure, with longer exposures required for lower pressure environments to reach equilibrium. Figure 6 shows similar heating of the RS92 and RS41 temperature sensors at 300 hPa (0.15 K), but at 3 hPa (Figure 7) the RS41 (1.4 K) heats only half as much as the RS92 (2.8 K). Furthermore, at 3 hPa, the RS41 sensor has reached equilibrium after approximately 30 s of illumination, whereas after 4 minutes of illumination the RS92 still hasn't reached equilibrium. The investigation of the RS41's radiation errors continues, including experiments with the newly developed laboratory radiation system, and the full analysis of these tests will be reported in a separate paper.

**6 Metrology**

A fundamental metrological principle stipulates that replacing one operational instrument with another should pose no problem provided that the results from both instruments are fully traceable to SI standards. Consequently, the new instrument could (almost) instantaneously be included in the traceability chain without the need for parallel testing or comparison with the replaced device. In practice, this idealized concept can rarely be adopted, even in primary metrology laboratories or in National Metrology Institutes. The problem is that different instruments or sensors may show different responses to external environmental factors.

Concerning radiosondes, the sensors, especially for humidity and temperature, may  be exposed to unavoidable atmospheric or ascent-related effects during a sounding. Some of these effects cannot  be completely duplicated during the controlled laboratory conditions which are provided during the preceding metrological instrument characterization and calibration procedures. Consequently, differences in the responses of sensors from different sonde models may still exist during ascents. An example is the warm bias of the radiosonde's temperature sensor caused by solar radiation.

For this reason it is essential to identify and quantify these differences between the old and the new measurement system at each time of a sonde replacement, not only by laboratory work, but also

by comparison flights where the instruments are carried bythe same balloon. The same should also be done when there have been significant changes in the design, sensor technology, or operation of an actual instrument.

400 An adequate characterization of the measurement uncertainties for entire radiosonde profiles includes the following components:

- Calibration uncertainty, given by the manufacturer (see product data sheets for Vaisala RS92 and RS41 available the manufacturer's web site, https://www.vaisala.com). The sensors are calibrated in Vaisala's CAL4 calibration facility (Vaisala, 2002) that contains PTU (pressure, temperature, and humidity) reference sensors, routinely re-calibrated against National Institute of Science and Technology of the United States of America (NIST)-traceable standards (for pressure and temperature) and the Finnish Centre for Metrology and Accreditation (Mittatekniikan keskus – MIKES, which has become part of VTT Technical Research Centre of Finland Ltd and is now officially known as VTT-MIKES) for humidity. Note that for GRUAN the uncertainties of the calibration curves are important, not so much the information about the overall uncertainties in soundings which are related to the manufacturer-provided data product,

- Uncertainties from GRUAN-prescribed pre-flight ground check procedures,

- Uncertainties estimated during GRUAN post-flight processing, including uncertainties of corrections to the measured raw data for systematic effects which are mainly derived from laboratory tests.

Differences revealed by instrument comparison flights indicate that some systematic effects have not yet been correctly assessed or even identified. To assure consistency of the measurement results of different instruments, the differences are to be evaluated in terms

420 of possible corrections, or, if this is not possible, the uncertainty budget is to be widened properly to account for them. It is important to arrange the comparisons network-wide in a coordinated manner, covering different seasons and locations. This is to ensure that This is to ensure that all possible sources of systematic errors such as latitude, climate, environmental and technical conditions be-

425 fore and during launch, local specifics in the sounding procedures or setups are tested during twin soundings.

[revised manuscript text omitted]

The National Oceanic and Atmospheric Administration (NOAA) Products Validation System (NPROVS) (Reale et al., 2012) facility _routinely_ associates satellite measurements with GRUAN _and operational_ observation data within a given _spatiotemporal_ collocation window. This system

480 can also process _associated output from NWP models and_ profile data from coincident ancillary measurements, including the_ir_ associated uncertainties. This _facilitates_ redundancy testing (Immler et al., 2010) and integration of consistent ancillary profiles into Site Atmospheric State Best Estimates (SASBE) (Tobin et al., 2006) , _which is another objective of GRUAN_. However, the processing, packaging and _integration of satellite,_

485 _radiosonde, and_ ancillary data in a spatially and temporally coherent manner is a complicated task that remains under discussion.

490

**7.2 Scheduling**

In addition to ground-based ancillary measurements such as GNSS, lidar, MWR, and FTIR,  _space borne_ observations _from polar orbiting radiometer sounders and satellite-based GNSS_

495 _radio occultation (GNSS-RO)_ present a valuable and abundant source of _redundant_ observations for comparisons with radiosounding data. To maximize their potential for scientific exploitation, the radiosonde launches should be scheduled to be coincident with satellite overpasses and/or the occurrence of  GNSS-RO over the site.

500

_Since the early 2000's, NOAA cooperates with the U.S. Department of Energy Atmospheric Radiation Measurement (ARM) program to perform radiosoundings that are collocated with satellite observations._A joint service between GRUAN and European Organisation for the Exploitation of

Meteorological Satellites (EUMETSAT) provides predictions of these overpasses, including so-called "golden overpasses" where the measurements of a polar orbiter, such as MetOp-A or MetOp-B, are spatially (distance $< 200\,\mathrm{km}$) and temporally (within 30 minutes) collocated with a GNSS-RO measurement. In addition, at GRUAN sites where ground-based ancillary measurements are also being made, there could be multiple redundant observations of target ECVs to enable better exploitation; see Section 7.1 for further discussion.

In the most basic context, satellites provide a consistent background or traveling calibration standard on a global scale to further interpret twin radiosonde results at a given site. Potential benefits include the identification of possible site bias and  in case of twin soundings, a measure of the performance of the radiosondes in various atmospheric environments. The additional information from satellite-synchronized launches can potentially minimize the number of twin launches required, reducing costs to the sites and maximizing opportunities for subsequent exploitation by the broader science community for myriad applications on a global scale.

[revised manuscript text omitted]